# NOISE-ROBUST DE-DUPLICATION AT SCALE

**Emily Silcock[1], Luca D'Amico-Wong[2], Jinglin Yang[3], Melissa Dell[4*]**

[1]Department of Economics, Harvard University; Cambridge, MA, USA.
[2]Harvard College; Cambridge, MA, USA.
[3]Department of Economics, University of California Berkeley; Berkeley, CA, USA.
[4]Department of Economics, Harvard University and NBER; Cambridge, MA, USA.
[*]Corresponding author: melissadell@fas.harvard.edu.

## ABSTRACT

Identifying near duplicates within large, noisy text corpora has a myriad of applications that range from de-duplicating training datasets, reducing privacy risk, and evaluating test set leakage, to identifying reproduced news articles and literature within large corpora. Across these diverse applications, the overwhelming majority of work relies on $N$-grams. Limited efforts have been made to evaluate how well $N$-gram methods perform, in part because it is unclear how one could create an unbiased evaluation dataset for a massive corpus. This study uses the unique timeliness of historical news wires to create a 27,210 document dataset, with 122,876 positive duplicate pairs, for studying noise-robust de-duplication. The time-sensitivity of news makes comprehensive hand labelling feasible - despite the massive overall size of the corpus - as duplicates occur within a narrow date range. The study then develops and evaluates a range of de-duplication methods: hashing and $N$-gram overlap (which predominate in the literature), a contrastively trained bi-encoder, and a "re-rank" style approach combining a bi- and cross-encoder. The neural approaches significantly outperform hashing and $N$-gram overlap. We show that the bi-encoder scales well, de-duplicating a 10 million article corpus on a single GPU card in a matter of hours. We also apply our pre-trained model to the RealNews and patent portions of C4 (Colossal Clean Crawled Corpus), illustrating that a neural approach can identify many near duplicates missed by hashing, in the presence of various types of noise. The public release of our NEWS-COPY de-duplication dataset, codebase, and the pre-trained models will facilitate further research and applications.

## 1 INTRODUCTION

Robust identification of near-duplicate texts in large, noisy corpora is important for a variety of applications. Duplication in training data degrades model performance (Lee et al., 2021), can raise serious privacy risks (Kandpal et al., 2022), and can degrade performance on downstream tasks (Schofield et al., 2017; Liu et al., 2022; Allamanis, 2019). Additionally, the presence of test set leakage complicates evaluation of model performance, concerns that are elevated with large language models that have greater capacity to memorize training data or can consult an external database. Patterns of duplication are also themselves of interest, for studying the dissemination of reproduced content such as literature or news (Cordell, 2015; Smith et al., 2015; Vesanto et al., 2017) and for reducing noise in datasets used for statistical analyses.

In contrast to the literature on semantic textual similarity, where deep neural architectures predominate - e.g. Reimers & Gurevych (2019) - text de-duplication overwhelmingly uses $N$-gram methods. There have been few efforts to formally evaluate the adequacy of $N$-gram based de-duplication or to explore potential performance gains from neural text de-duplication. This study builds a large de-duplication dataset and develops neural methods for robust textual de-duplication that significantly outperform $N$-gram based methods and scale efficiently.

A major hurdle to overcome in systematically studying text de-duplication is the lack of data for an unbiased evaluation of different methods. Typically, there is no way to exhaustively identify all du-

plicates of a given example in a large corpus, complicating comparisons of recall. To circumvent this challenge, we examine duplication in historical news. Reproduction from news wires and syndicate services was widespread, forming over half the content of U.S. local newspapers. Media historian Julia Guarneri (2017) writes: "by the 1910s and 1920s, most of the articles that Americans read in their local papers had either been bought or sold on the national news market... This constructed a broadly understood American 'way of life' that would become a touchstone of U.S. domestic politics and international relations throughout the twentieth century." Because news is timely, reproduction happens within a narrow time window, and hence annotators can exhaustively identify all duplicates despite the massive overall size of the corpus. To build an unbiased evaluation sample, highly skilled human annotators manually reviewed every front page article from 973 newspapers on four randomly chosen days in 1930, 1955, and 1974 to create clusters of duplicated articles (including all singletons). Additional data, spanning the period from 1920 to 1977, were compiled for model training. The resulting public `NEWS-COPY` dataset - which contains 27,210 articles, comprising 122,876 positive duplicate pairs - aims to encourage further study of robust de-duplication.

In the absence of evaluation data, the literature has largely assumed that text de-duplication is sufficiently simple that neural methods are not required. However, noise is an integral feature of large text datasets, resulting from OCR errors, abridgement, news aggregators, plagiarism, or machine translation, to name a few reasons. This can lead near duplicate documents to have low $N$-gram similarity. Amongst duplicated pairs of articles in the `NEWS-COPY` test set, the average Jaccard similarity using 3-grams (4-grams, 5-grams) between pairs of reproduced articles is 30% (26%, 23%). 19% of duplicates have no 10-grams in common and 31% have no 15-grams in common, often as a result of minor text noise. Neural methods are plausibly more robust.

Using the `NEWS-COPY` dataset, we examine different text de-duplication methods that vary along two key dimensions: whether or not the method is neural and computational cost. Drawing inspiration from work on semantic textual similarity and on retrieval, we develop two approaches for neural text de-duplication: a contrastively trained bi-encoder plus clustering method and a 'reranking' style method, which uses a computationally cheap transformer bi-encoder to measure the pairwise similarity between all articles and then passes each article's nearest neighbors to a cross-encoder, at an additional computational cost. We also examine $N$-gram overlap and locally sensitive hashing, the latter of which is highly scalable. The neural methods significantly outperform the non-neural approaches. The Adjusted Rand Index (ARI) for the re-rank model is 93.7 and for the bi-encoder model is 91.5, versus 73.7 for LSH and 75.0 for $N$-gram overlap.

While the primary advantage of hashing - and a central motivation for its frequent usage - is its scalability, massive scale similarity search (Johnson et al., 2019) is sufficiently cheap on modern GPUs to make neural de-duplication highly scalable. We use our contrastively-trained bi-encoder and a single NVIDIA 40GB A6000 GPU card to de-duplicate a 10 million document, 19 GB corpus in 11 hours and 45 minutes. While this cost is already marginal in the context of working with large text corpora, it could be reduced significantly further by using a lighter weight language model, as the majority of the time cost is embedding the 10M articles.

The publicly available neural de-duplication models, available at `https://github.com/ dell-research-harvard/NEWS-COPY`, can be applied to novel de-duplication problems. To evaluate off-the-shelf performance, we apply our bi-encoder model to two subsets of C4 (Colossal Clean Crawled Corpus), a massive dataset created by applying a series of filters to a single snapshot of Common Crawl (Raffel et al., 2019; Dodge et al., 2021): RealNews - which consists of around 13 million digital news articles - and all 90,671 patents scraped from Google's online patent database. We also examine test set leakage between SuperGlue (Sarlin et al., 2020) and RealNews. While there is not an unbiased ground truth measure for these datasets, an analysis of predicted duplicates shows that the bi-encoder detects a variety of noisy duplicates that hashing overlooks, which result from aggregators of digital news, machine translation, and other sources of noise.

The rest of this paper is organized as follows: Section 2 provides an overview of the relevant literature. Section 3 describes the `NEWS-COPY` dataset, and Section 4 develops neural de-duplication methods and their non-neural comparisons. Section 5 evaluates the performance of different de-duplication methods, Section 6 explores scaling, and Section 7 applies de-duplication to a subset of C4. Finally, Section 8 concludes.

## 2 LITERATURE

**De-Duplication:** Textual de-duplication is a fundamental task for curating the large text corpora that support the deep learning revolution. Lee et al. (2021) review the de-duplication literature, providing evidence that duplication in training datasets is widespread: e.g. Dodge et al. (2021) find up to 14.4% of test examples of various standard benchmarks verbatim in C4 and Bandy & Vincent (2021) document that the Books Corpus (Zhu et al., 2015) - used in training BERT (Devlin et al., 2018), GPT (Brown et al., 2020), and other large language models - contains 4,255 unique books and 2,930 books that are exactly duplicated at least once.

Lee et al. (2021) document that models trained on deduplicated data regenerate approximately 10 times less training data, and Kandpal et al. (2022) find a superlinear relationship between the number of times a sequence is present in training data and regeneration, with a sequence present 10 times being regenerated 1000 times more often than a sequence present once. Carlini et al. (2022) find that the likelihood of a model generating exact continuations from the training data scales with model size, training data duplicates, and prefix length. This could raise plagiarism risks (Lee et al., 2022).

There is also a literature showing that duplicates adversely affect downstream tasks. Schofield et al. (2017) study the impact of text duplication on semantic models, documenting that substantial over-representation can overwhelm meaningful topical patterns. Allamanis (2019) show that duplication in code datasets worsens performance on code understanding. Liu et al. (2022) show that de-duplication of an open electronic health record database significantly improves clinical natural language processing models. Moreover, when training LMs that can consult a massive external database - as in a retrieval enhanced transformer language setup (Borgeaud et al., 2022) - test set leakage becomes a particularly salient concern. Borgeaud et al. (2022) conclude: "Further work is yet needed to better understand the role of test set leakage in the performance of LMs."

Non-neural methods predominate in textual de-duplication (Leskovec et al., 2020). Borgeaud et al. (2022) compute 13-gram Jaccard similarity between train and test documents using MinHashing and remove all training documents with 0.8 similarity or higher to validation/test documents. Radford et al. (2019) use 8-gram overlaps for post-hoc identification of duplication between GPT-2's training data and evaluation datasets, and Brown et al. (2020) remove from the GPT-3 training data any example with a 13-gram overlap with an evaluation example. Other de-duplication contexts include large datasets of medical notes (Shenoy et al., 2017) and scholarly articles (which can include updates) (Gyawali et al., 2020), both of which have been examined with locally sensitive hashing.

Identifying reproduced texts within historical newspapers is itself an application that has generated considerable interest. The Viral Texts Project (Cordell, 2015; Smith et al., 2015) uses $N$-gram comparisons to track the dissemination of reproduced literature in antebellum newspapers. Viral Texts utilizes the Chronicling America (Culpepper, 2007) OCR, which does not recognize individual articles, headlines, captions, etc. This leads to scrambled up texts. We first apply object detection methods to the document layouts (He et al., 2017; Shen et al., 2021) to extract structured texts of individual articles that allow us to capture performance gains from the language understanding of neural methods.

Vesanto et al. (2017) use NCBI BLAST, a software for comparing and aligning biological sequences, to quantify text reproduction at scale in Finish newspapers from 1771 to 1910. They remove all characters besides the 23 most common letters from an uncased corpus of Finish newspapers, and then convert these to the alphabet of 23 amino acids recognized by BLAST. BLAST is used to make pairwise comparisons between all documents in the corpus, indicating which pairs have text overlap. To scale the problem, we use hashing - which avoids the need to convert texts into amino acid sequences - or a contrastively trained bi-encoder - which leverages the power of deep learning.

**Semantic Textual Similarity:** There are important parallels between semantic textual similarity (STS) and textual de-duplication. Notably, our bi-encoder method draws inspiration from Sentence BERT (S-BERT) (Reimers & Gurevych, 2019), and we use an S-BERT pre-trained bi-encoder as our base language model. S-BERT adds a pooling operation to BERT/RoBERTa embeddings - that takes the mean of all output vectors - to derive a fixed sized sentence embedding that can then be examined with clustering methods.

**Retrieval**: We draw inspiration for our reranking approach from the literature on open domain retrieval and question answering (Wang et al., 2018; Lin et al., 2018; Karpukhin et al., 2020; Thakur

et al., 2021; Wu et al., 2019), which avoids the infeasible quadratic cost of applying a cross-encoder to a massive corpus by first ranking documents with a bi-encoder (or with sparse methods). In our re-ranking model, instead of a passage encoder and a query encoder, there is a symmetric bi-encoder.

## 3 THE `NEWS-COPY` DATASET

### 3.1 REPRODUCTION IN NEWS

Reproduction is an important feature of news. News wire services distribute stories written by their own news bureaus and by member newspapers to member news outlets, whereas syndicates disseminate to their subscribers columns written by freelance journalists or purchased from newspapers. The nation's largest newspapers also ran syndicate services to redistribute their own stories. The main news wire services in the United States historically were the Associated Press (AP), the United Press (UP), and the International News Service (INS), the latter two of which merged to form United Press International (UPI) in 1958.

Editing could take place at multiple places along the news transmission chain. Wire staff verified and edited stories after receiving them from members, and then stories could be edited again by local wire bureaus, of which there were around 100 for the Associated Press. Finally, local newspapers could abridge content to fit space requirements. This leads to a range of near duplicates in the presence of abridgement and OCR noise. Noisy duplicates in news are not limited to the historical context, with digital news aggregators today leading to a similar phenomenon (Coddington, 2019).

### 3.2 DESCRIPTION OF THE `NEWS-COPY` DATASET

Table 1 summarizes the key features of the `NEWS-COPY` dataset. It consists of 27,210 articles, drawn from 973 newspapers between 1920 and 1977.[1] `NEWS-COPY` contains two types of data: data for training and four full day exhaustively labeled evaluation samples, constructed with two consecutive days of content in 1930 and single days in 1955 and 1974, selected at random. The 1955 sample is a validation set used to select hyperparemters for both the $N$-gram and neural methods. 1930 and 1974 are pooled to form the test set and used only to produce the results shown in this paper. In the full day samples, there are far more negative than positive pairs, as is generally the case in de-duplication problems, whereas the training data contain a more balanced sample.

### 3.3 PROCEDURE FOR BUILDING THE DATASET

To build `NEWS-COPY`, we first apply Layout Parser (Shen et al., 2021) with a custom-trained object detection model (He et al., 2017) to front page scans of off-copyright historical newspapers to identify individual article bounding boxes. The contents of article bounding boxes are OCR'ed with Tesseract. When component bounding boxes span multiple columns on the same page, the OCR'ed texts are associated into full articles using a rule-based association method that exploits the coordinates of headline and article bounding boxes. This pipeline extracts the structured article texts. Headlines were chosen by local newspapers - not wires - and as a result are rarely reproduced and not included in the dataset. Weather forecasts are removed by running a distil-RoBERTa classifier trained on 392 labeled articles (179 positive, 202 negative). This removes 4.4% of the validation set and 3.3% of the test set. We also hand-removed documents containing incorrectly merged article bounding boxes from different underlying source articles (as there was no single ground truth cluster to which these articles belonged), and news summaries, which summarize multiple news stories in a single article and hence also have no clear cluster with which they are associated. These represent 3.4% and 3.3% of the validation and test sets, respectively.

Duplicates are defined as articles that came from the same original source article, regardless of the degree of abridgement or OCR noise. Articles from different source articles that contain the same quote are labeled as non-duplicated. Likewise, articles updated to reflect breaking news are labeled as different, as are different articles on the same overarching story.

---

[1]A copyright law change effective January 1, 1978 resulted in nearly all newspapers from that date forward being under copyright by default.

|  | Positives Pairs | Negative Pairs | Reproduced Articles | Singleton Articles | Total Articles |
|---|---|---|---|---|---|
| **Training Data** | | | | | |
| Training | 36,291 | 37,637 | 891 | – | 7,728 |
| Validation | 3,042 | 3,246 | 20 | – | 283 |
| **Full Day Evaluation** | | | | | |
| Validation | 28,547 | 12,409,031 | 447 | 2,162 | 4,988 |
| Test | 54,996 | 100,914,159 | 1,236 | 8,046 | 14,211 |
| **Full Dataset** | 122,876 | 113,364,073 | 2,594 | 10,208 | 27,210 |

Table 1: This table provides summary statistics from the `NEWS-COPY` dataset, decomposed into the training sample and the full day evaluation data.

To construct the full-day samples, we first ran $5$-gram overlap with a very conservative $N$-gram overlap threshold of 1% to create large candidate duplicate clusters. Highly trained student research assistants carefully reviewed these clusters, breaking false positive links. A sub-sample was doubled-labeled to ensure our definition of a duplicated article was coherent, and that labeling was consistent across annotators. Interannotator agreement on a subset of 8512 pairs was 98.1% (90.9 Cohen's Kappa). Next, annotators reviewed each of the resulting clusters to merge together clusters as needed. Finally, annotators exhaustively reviewed every singleton article, associating them with article clusters as needed. Articles were sorted by byline (recognized with a custom-trained named entity recognition model) to facilitate this process. For building the training data, the approach was similar, which provides hard negatives. We did not review all singletons, as the aim was to produce labeled batches for constrastive training. About two thirds of the negative pairs in the training data are hard negatives, with the remaining third coming from randomly selected article pairs.

## 4 MODEL ARCHITECTURES

### 4.1 THE BI-ENCODER MODEL

We contrastively train a symmetric bi-encoder to learn similar representations for near duplicate articles and dissimilar representations for non-duplicated articles. We use an S-BERT MPNET model (Reimers & Gurevych, 2019; Song et al., 2020) contrastively trained on over a billion sentence pairs - drawn from STS datasets - as the base language model. The S-BERT architecture pools representations for up to the first 512 tokens in each article, using mean pooling, to construct a document level representation. Like Reimers & Gurevych (2019), we found when experimenting with vanilla RoBERTa embeddings - which also perform well on de-duplication - that mean pooling of each of the representations significantly outperforms using the [CLS] token to represent the document. S-BERT provides a speed-optimized implementation of this pooling strategy. We chose the MPNET S-BERT because it performs best overall on STS benchmarks.

We use S-BERT's online contrastive loss (Hadsell et al., 2006) implementation, with a 0.2 margin and cosine similarity distance. The learning rate is 2e-5 with 100% warm up and a batch size of 32. We use an AdamW optimizer, and the model is trained for 16 epochs.

The bi-encoder dense document representations can be clustered to identify duplicates. We use FAISS (Johnson et al., 2019) to compute all embeddings within a given distance range, a hyperparameter tuned on the full-day validation sample. This output is used to build a graph, where nodes are articles and edges connect articles within the threshold distance. Connected components can be extracted to define clusters - which is equivalent to single linkage clustering - or Louvain community detection can be applied to the graph to control false positive edges that can merge otherwise disparate groups of articles.

### 4.2 THE RE-RANKING MODEL

While a cross-encoder can offer the most flexible, expressive comparisons between texts, it requires $N^2$ embeddings to compare $N$ texts, infeasible in large corpora. To make the use of a cross-encoder feasible, we draw inspiration from the retrieval literature (Wang et al., 2018; Lin et al.,

|  | **Neural** | **Non-Neural** |
|---|---|---|
| **Most scalable** | Bi-encoder (91.5) | LSH (73.7) |
| **Less scalable** | Re-ranking (**93.7**) | $N$-gram overlap (75.0) |

Table 2: The numbers in parentheses are the Adjusted Rand Index for four different models - a bi-encoder, a "re-ranking" strategy that combines a bi- and cross-encoder, locally sensitive hashing (LSH), and $N$-gram overlap. Hyperparameters were chosen on the NEWS-COPY validation set, and all models were evaluated on the NEWS-COPY test set.

2018; Karpukhin et al., 2020; Thakur et al., 2021; Wu et al., 2019) by first ranking document similarity with a bi-encoder, and then passing the most similar documents to a cross-encoder. This approach, while not as cheap as simply clustering the bi-encoder embeddings, can still scale to a significant extent. We test whether it offers additional performance gains over using a bi-encoder alone. For the baseline re-ranking model, we choose a bi-encoder threshold of 0.92, optimized using the one-day validation sample (the supplementary material examines robustness to this threshold). We use RoBERTa-base (Liu et al., 2019) as the base language model, with a 2e-5 learning rate and an AdamW optimizer. It is trained for 5 epochs with 20% warmup and a batch size of 32.

## 4.3 N-GRAM METHODS

We explore two different $N$-gram methods: locality-sensitive hashing (LSH), as well as one which relies on full computation of $N$-gram overlaps. For both methods, we first shingle each article into $N$-grams, where $N \in \{3, 4, 5, 10, 15\}$. Punctuation is removed from the text of the article during pre-processing to reduce noise. We define overlap as the Jaccard similarity between two documents, given by $\frac{|A \cap B|}{|A \cup B|}$. We compute overlaps between each pair of articles, drawing an edge between two articles if the overlap is above some minimum overlap threshold. This hyperparameter is tuned on the validation sample. We also choose $N$ using the validation sample, with $N = 3$ providing the best Adjusted Rand Index. Once all edges have been drawn, Louvain community detection can be applied to remove false positives.

For LSH, we use Datasketch's MinHashLSH library, which is widely used in the literature. The two relevant hyperparameters are the number of hash functions and the minimum number of hash collisions needed to designate two documents as duplicates. For the former, we choose 10, which predominates in the literature, balancing computational efficiency and accuracy. The latter is tuned on the validation sample. All documents with at least the threshold number of hash collisions are defined as duplicates. As with the other methods, community detection can be run once the graph has been constructed.

## 5 MODEL EVALUATION

### 5.1 BASELINE RESULTS

Table 2 compares the accuracy of 4 baseline models on the full-day test samples: non-neural methods ($N$-gram overlap and hashing) and neural methods (bi-encoder and re-ranking). For both of these, we examine a scalable method (hashing and bi-encoder) and a method that aims to achieve higher accuracy at some computational cost ($N$-gram overlap and re-ranking). We do not evaluate the cross-encoder alone, as the scale of most de-duplication applications makes it computationally infeasible.

The neural methods significantly outperform the $N$-gram based methods, increasing the adjusted Rand index (ARI) (Hubert & Arabie, 1985) from 73.7 to 91.5 when comparing the most scalable methods (LSH and bi-encoder) and from 75.0 to 93.7 when comparing the more computationally costly methods ($N$-gram overlap and re-ranking).[2] In contrast, the more computationally intensive methods offer little advantage over their more scalable counterparts. The 're-ranking' strategy does offer modest gains over using a bi-encoder alone (ARI of 93.7 vs. 91.5), making it a compelling method for moderately size datasets where accuracy is paramount.

---

[2]Evaluating with pairwise F1 leads to similar conclusions.

These results underscore the potential returns of applying deep neural models to de-duplication of massive text datasets. When neural false positives occur, it is typically in the context of articles that have some repeated content but do not meet the definition of duplicates used to create the `NEWS-COPY` dataset: e.g. the articles contain the same quote, are different articles about the same story, or one is an updated article that contains additional breaking news. False negatives are most likely to occur when OCR errors are severe or abridgement is severe. If desired, further improvements in accuracy could plausibly be achieved by adding more of these challenging edge cases to the training data. A quantitative analysis of the errors is given in the Appendix.

With $N$-gram methods, errors are mechanical, occurring in articles where noise results in fewer $N$-grams in common or where distinct articles by chance have significant overlap. $N$-gram overlap and hashing control this tradeoff through a threshold, whereas neural methods are highly flexible and can be finely tuned to the downstream task through curation of appropriate training data.

The Appendix examines a variety of ablations. The neural methods are robust to variations such as changing the contrastive loss function, changing the bi-encoder clustering threshold, and changing the clustering method (hierarchical agglomerative clustering slightly outperforms the baseline, at the cost of being less scalable). As expected, off-the-shelf S-BERT (designed for *semantic* textual similarity) underperforms our textual de-duplication model (ARI 70 for the bi-encoder and 69.3 for re-ranking). For the non-neural methods, changing the $N$ used to construct $N$-grams also leaves the broad findings unchanged, as does using character level $N$-grams and forcing all text to its nearest dictionary match. In all cases, when we split the test sample into articles from 1930 and 1974 - running analyses separately - the results are also qualitatively unchanged.

## 6  COMPUTATIONAL EFFICIENCY

Hashing is often advocated because of its scalability, and neural methods for de-duplicating large text corpora likewise need to be highly scalable. We conduct experiments on a 19 GB, 10 million article corpus, created by applying the same object detection model used to curate `NEWS-COPY` to millions of front page newspaper page scans. These experiments use a 32-Core 3.50 GHz AMD RYZEN Threadripper Pro 3975WX and a single NVIDIA A6000 GPU, a very modest setup for working with large text corpora.

We scale the bi-encoder and LSH approaches to this corpus, reporting speeds in Table 3. The largest cost of scaling the bi-encoder, by a wide margin, is embedding the 10M articles, which takes 8:38:52 on a single GPU. This could be sped up significantly by deploying a smaller language model. However, since this cost is already fairly marginal in the context of working with massive text datasets, we do not explore this here.

FAISS (Johnson et al., 2019), with `IndexFlatIP`, is used to calculate pairwise exact distances between the embeddings.[3] These computations require just over 3 hours on a single GPU. This produces a list of article pairs whose embeddings are within a threshold distance, using the optimal threshold selected in the NEWS-COPY full day validation sample. We build a graph - where each of the 10M documents is a node and edges connect documents whose embeddings are within the threshold distance. This takes 0:01:23 using `CuDF` and `CuGraph`. False positives are controlled by running Louvain community detection on the graph; alternatively, one could define clusters simply by extracting the connected components. Total run-time on the single GPU card is 11:45:10.

While neural methods are not expected to be faster than hashing, they compare reasonably. LSH requires 3:39:05 CPU hours for pre-processing, shingling, and hashing, and around 1 GPU minute to create the resulting graph and apply community detection. Commonly used hashing libraries run on a CPU, and hence this is where we implement LSH, with Datasketch's MinHashLSH. To effectively scale LSH, we slightly modify the previous architecture and break the hashes into bands comprised of rows of hashes. Each of these bands is hashed again and two articles that share an identical band hash are considered duplicates. The choice of bands and rows determines an $S$-curve,

---

[3]Because FAISS range search is not GPU-supported, we implement $k$ nearest neighbor search, conservatively setting $k$ to 900. Then distances are filtered to those below the optimal threshold found in our one day validation sample. In NEWS-COPY, articles are never reproduced more than 200 times, with the average reproduced article appearing 6.3 times, indicating that $k = 900$ is quite conservative.

|  | Embed
Articles | Compute
Similarity | Build
Graph | Commun.
Detect. | Total
Time | Mean times
Reproduced |
|---|---|---|---|---|---|---|
| Bi-Encoder | 8:38:52
(GPU) | 3:04:53
(GPU) | 0:01:23
(GPU) | 0:00:02
(GPU) | 11:45:10
(GPU) | 6.41 |
| Hashing |  | 3:39:05
(CPU) | 0:00:55
(GPU) | 0:00:08
(GPU) | 3:40:08
(mostly CPU) | 11.55 |

Table 3: This table reports computational efficiency in scaling the bi-encoder and LSH methods to a 10 million article corpus. Parentheses indicate whether the calculations were run on a CPU or a single NVIDIA A6000 GPU card. *Mean times reproduced* reports the average size of duplicated article communities that each method estimates.

which gives the probability of being considered a duplicate given a certain Jaccard similarity. To generate our desired $S$-curve, we used 15 bands and 2 rows.

The bi-encoder method detects 810,080 distinct reproduced articles. The average duplicated article was reproduced 6.41 times, strikingly similar to what we document in the single day `NEWS-COPY` labeled data. This is expected, since most reproduced articles occur within the time window captured in the `NEWS-COPY` full day samples. The number of true negative pairs is massively greater in the 10M article corpus, and it is encouraging that the average number of times articles are estimated to be reproduced remains quite constant, rather than being blown up by false positives. In contrast, LSH detects only 486,460 distinct reproduced articles, estimating that each on average appears 11.55 times. This is significantly greater than in the labeled `NEWS-COPY` data, and likely indicates that false positives are significantly increasing the size of detected duplicated article groups.

## 7    NOISY DUPLICATES IN COMMON CRAWL

To evaluate off-the-shelf performance in the presence of varying types of noise, we apply our pre-trained bi-encoder model and hashing to two subsets of C4 (Colossal Clean Crawled Corpus), a massive dataset created by applying a series of filters to a single snapshot of Common Crawl (Raffel et al., 2019; Dodge et al., 2021): RealNews - which consists of around 13 million digital news articles - and all 90,671 patents from Google's online patent database, which is the largest single domain in C4. While the ground truth for duplication in these datasets is unknown, an analysis of predicted duplicates suggests that the neural method detects a variety of noisy duplicates that a standard application of hashing overlooks.

In the RealNews data, news aggregators, which tend to make small edits to digital news stories before releasing them on their own sites (Coddington, 2019), can generate noisy duplicates. In the patents data, one source of duplicates is noisy translation, as patents are routinely filed in multiple countries and non-English language patents were machine-translated into English. OCR noise is also present for older patents. A de-duplicated Google patent dataset could be of direct relevance to researchers working with patent data, which are the backbone of a large literature studying the drivers of innovation.

|  |  | Clusters | Art. in
Cluster | Cluster
Size | Unique
Clusters | Unique
Dups. | False
Positives |
|---|---|---|---|---|---|---|---|
| **RealNews** | Neural | 323,913 | 902,019 | 2.8 | 307,080 | 558,710 | 20% |
|  | Hashing | 30,844 | 81,994 | 2.7 | 7,195 | 20,308 | 44% |
| **Patents** | Neural | 2,326 | 7,431 | 3.2 | 2,300 | 5,078 | 4% |
|  | Hashing | 339 | 733 | 2.2 | 300 | 355 | 16% |

Table 4: Statistics for training set de-duplication are reported for the bi-encoder and hashing. 'Unique clusters/duplicates' reports the number of duplicate clusters/articles that are predicted by each model but not the other. False positive are calculated by looking at a sample of duplicates that are uniquely predicted by each model.

The bi-encoder is applied off-the-shelf, with the same hyperparameters as used in the `NEWS-COPY` analyses. We also use the same definition of a noisy duplicate. For hashing, we found that applying

the same parameters resulted in a large number of false positives, detecting 40 times more duplicate pairs than the bi-encoder. This further underscores the brittleness of rule-based methods. Instead, we follow the approach of Borgeaud et al. (2022) for controlling test set leakage in the Retrieval Enhanced Transformer model: hashing with 10-gram collisions and a Jaccard similarity threshold of 0.8. Following GPT-3 (Radford et al., 2019), we ignore clusters above a certain size (using a threshold of 1000 for RealNews and 300 for patents). The bi-encoder detects 27 of these for RealNews and 7 for patents, whereas they do not appear with hashing. These tend to be templates for articles such as medical reports, crime reports, and obituaries. Randomly-selected examples are shown in the Appendix. These might also be undesirable for training and downstream tasks, but we ignore them to be conservative.

Table 4 reports statistics from de-duplicating the training set of RealNews, as well as all patents (which does not have a train-test split). It reports several measures of duplicates: the number of clusters of duplicated texts, the total number of articles in a duplicate cluster, average cluster size, the number of unique clusters detected with one method but not the other, and the number of texts marked for removal with one method but not the other. The neural method identifies around an order of magnitude more duplicates than hashing. Average cluster size is roughly similar. The Appendix provides randomly selected examples of predicted duplicate clusters. The duplicates predicted by the neural method but not hashing are true positives with reasonably high probability. In a random sample of 50 clusters that are detected by the neural method but not by hashing, 20% are false positives for RealNews and 4% are false positives for the patent data. As in NEWS-COPY, many of the false positives consist of updated news stories. When examining duplicates predicted by LSH but not the bi-encoder, 44% are false positives in RealNews and 16% are false positives for the patents.

Table 5 examines test set leakage from the RealNews training set to the RealNews development set and the SuperGlue benchmark (Sarlin et al., 2020).[4] For datasets in SuperGlue that do not primarily consist of news, test set leakage is low. These serve as a placebo, showing that neither method detects duplicates when there is no reason to expect many. In contrast, when examples are drawn from news, the neural method predicts considerable leakage. An analysis of a random sample of (up to) 50 neural predicted duplicates suggests that most are true positives, with the false positive rate ranging from 0% on 17 duplicates predicted from BoolQ (drawn from Wikipedia) to 16% on a random sample of 50 duplicates predicted from the RealNews development set. These false positives are often updated articles. In contrast, hashing finds almost no duplicates. The Appendix provides examples of predicted duplicate clusters. Further, we find that GPT-3, which is trained on RealNews, performs substantially better on samples from ReCoRD that have near duplicates in RealNews, compared to those that did not. More details on this experiment are given in the Appendix.

| Dataset | RealNews | BoolQ | CB | COPA | MultiRC | ReCoRD | RTE | WiC | WSC |
|---|---|---|---|---|---|---|---|---|---|
| Dev set size | 13863 | 3270 | 56 | 100 | 4848 | 10000 | 277 | 638 | 104 |
| Unique texts | 13863 | 2939 | 56 | 200 | 83 | 7132 | 277 | 1276 | 41 |
| Source | News | Wikipedia | Mixed | Constructed | Mixed | News | Mixed | Constructed | Constructed |
| Neural duplicates | 903 | 17 | 0 | 0 | 1 | 519 | 3 | 0 | 0 |
| LSH duplicates | 88 | 6 | 0 | 0 | 0 | 2 | 0 | 0 | 0 |
| Neural dup. false pos. | 16% | 0% | - | - | 0% | 3% | 0% | - | - |

Table 5: Columns report the dataset used. *Dev set size* is the number of examples in the development set for each dataset and *unique texts* is the number of texts that remain after pre-processing to remove exact duplicates. *Source* reports the dataset source. *Neural duplicates* is the number of duplicates found by the bi-encoder method and *LSH duplicates* is the number of duplicates found by hashing.

## 8 CONCLUSIONS

De-duplication is important to a range of NLP training and evaluation tasks, as well as for creating datasets that are of direct interest to researchers. This study provides evidence that neural methods offer significant performance gains - even on corpora that differ significantly from examples seen during training - and are highly scalable. These results suggest that neural text de-duplication is straightforward and well worth further examination in a range of contexts where it might improve model performance, control test set leakage, or yield cleaner datasets for downstream analyses.

---

[4] We compared every unique text in SuperGlue to the RealNews training set. In SuperGlue datasets with a premise and hypothesis we appended these together. WiC has two options for hypotheses, so we include both combinations, which is why the number of unique texts is larger than the size of the development set.

ACKNOWLEDGMENTS

We thank Abhishek Arora, Vania Cheung, and William Cox for excellent research assistance.

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

## A  APPENDIX

### A.1  ABLATIONS

Tables A1 - A4 explore several sets of ablations to the neural and non-neural methods. Community detection can detect false positive links between otherwise disparate clusters. Removing it makes little to no difference for the neural models (Table A1) but leads to a somewhat larger decline in performance for $N$-gram overlap and hashing (Table A3). This underscores the greater robustness of the neural approaches, whereas the non-neural methods are more prone to false positives that link otherwise disparate groups of articles.

Next, Table A1 considers whether hierarchical agglomerative clustering (HAC), which does not scale well, improves the accuracy of the bi-encoder approach. There are modest gains, with ARI increasing from 91.5 to 92.5, which highlights the potential returns of using HAC for moderately-sized de-duplication problems.

|  | Baseline | No commun. Detection | HAC Clustering | S-BERT STS | Loss Function | |
|---|---|---|---|---|---|---|
|  |  |  |  |  | SupCon | MNRL |
| Bi-encoder | 91.5 | 91.5 | **92.5** | 70.0 | 87.2 | 84.5 |
| Re-ranking | **93.7** | 92.0 | - | 69.3 | 91.1 | 89.9 |

Table A1: The first column removes community detection, and the second column uses hierarchical agglomerative clustering for the bi-encoder. *S-BERT STS* uses off-the-shelf S-BERT semantic textual similarity models. *SupCon* uses a supervised contrastive loss function and *MNRL* uses a multiple negatives ranking loss function.

Table A1 also considers off-the-shelf S-BERT (Reimers & Gurevych, 2019), a model trained to detect semantic textual similarity (STS). STS and de-duplication are distinct problems - despite STS being framed sometimes as the removal of (semantic) duplicates. As expected, the models designed for STS perform inadequately on textual de-duplication (ARI 70 for the bi-encoder and 69.3 for re-ranking), though these results are not that much worse than LSH despite not being trained for this task.

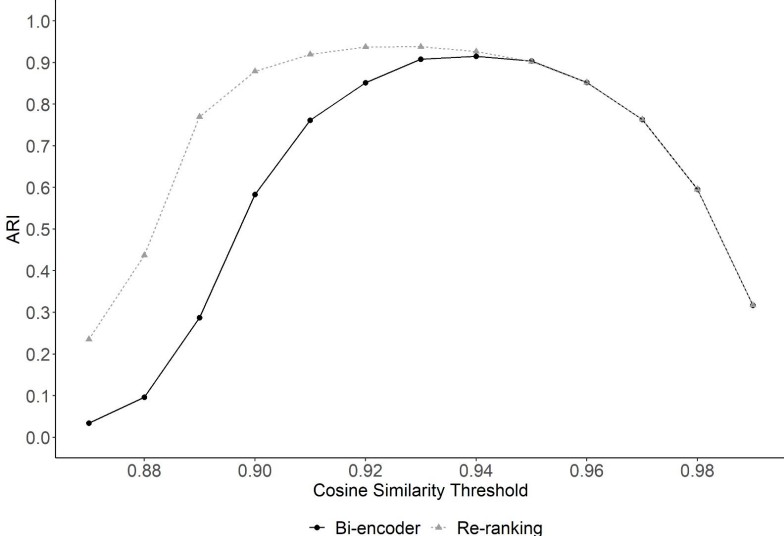

Figure A1: Model performance with different Cosine Similarity Thresholds

For the neural methods, we also explore different loss functions for contrastively training the bi-encoder - supervised contrastive (SupCon) loss (Khosla et al., 2020)[5] and multiple negatives ranking

---

[5]Trained for 32 epochs, 67% warm up, batch size 32, AdamW optimizer, 2e-5 learning rate.

loss (MNRL) (Henderson et al., 2017).[6] Online contrastive loss outperforms both of these alternative losses in both neural approaches. Moreover, Figure A1 examines robustness to varying the cosine similarity threshold, showing that results are quite robust.

An alternative to word level $N$-grams is character level $N$-grams. These have the downside of being substantially more computationally costly. Table A2 examines $N$ equal to 10, 15, 20, and 25. Performance is modestly better for hashing and modestly worse for $N$-gram overlap. Neural methods beat character $N$-grams by a wide margin.

|  | Baseline | 10-grams | 15-grams | 20-grams | 25-grams |
|---|---|---|---|---|---|
| Hashing | 73.7 | 78.2 | 76.3 | 77.2 | 71.3 |
| $N$-gram overlap | 75.0 | 68.1 | 68.4 | 72.3 | 73.9 |

Table A2: Numbers report the Adjusted Rand Index. The columns vary the $N$ used to compute character level $N$-grams.

To explore the potential of more aggressive text cleaning to improve the $N$-gram based methods, we allow SymSpell to correct words up to an edit distance of 5, which results in the overwhelming share of spell-corrected text appearing in the dictionary. This modestly improves performance for both hashing (ARI 75.9) and $N$-gram overlap (ARI 81.1), without changing the overall conclusions (Table A3). For the $N$-gram based methods, we chose the optimal $N = 3$ on the one-day validation sample. Table A3 also varies $N$ - with the overlap threshold/number of collisions for each $N$ again chosen to maximize ARI in the one-day validation sample. For both LSH, and $N$-gram overlap $N = 3$ remains optimal in the test set.

|  | Baseline | No commun. detection | In Dict. | 4-grams | 5-grams | 10-grams | 15-grams |
|---|---|---|---|---|---|---|---|
| Hashing | 73.7 | 67.7 | **75.9** | 71.5 | 69.6 | 63.7 | 59.4 |
| $N$-gram overlap | 75.0 | 70.2 | **81.1** | 69.0 | 72.3 | 63.5 | 60.9 |

Table A3: Numbers report the Adjusted Rand Index. *No community detection* removes community detection. In Dict. applies spell checking parameters that ensure that the overwhelming share of words appear in a dictionary. The other columns vary the $N$ used to compute $N$-grams and include community detection.

The baseline results pool the labeled full-day samples from 1930 and 1974. When these are analyzed separately, the average ARI between the two samples is similar to that in the pooled sample, as there are few false positive links between decades. Performance is better in the 1970s across methods due to a cleaner OCR.

|  | Biencoder | Re-ranking | LSH | $N$-gram overlap |
|---|---|---|---|---|
| 1930 | 86.1 | 90.4 | 57.8 | 44.7 |
| 1974 | 95.7 | 96.7 | 85.7 | 88.6 |

Table A4: This table separates the test set into articles from 1930 and articles from 1974.

## A.2 ERROR ANALYSIS

To understand the type of errors that were most commonly made by our bi-encoder method, we sampled 50 false positive pairs and 50 false negative pairs from the NEWS-COPY analysis and categorised them by types of error.

---

[6]Trained for 32 epochs, 0% warmup, batch size 16, AdamW optimizer, 2e-5 learning rate.

### A.2.1 FALSE POSITIVES

Amongst the 50 randomly selected false positive examined, 48 contained articles that were about the same story, but were either from different wire services, or different coverage from the same wire service. 2 were incorrectly labelled. The same errors were made repeatedly. For example, seven of the examples contained duplicates of the two articles about a fire that are in table A5.

| | |
|---|---|
| WASHINGTON (AP) — The House has passed legislation raising the minimum wage from $1.60 an hour to $2 this year for most workers covered and to $2.30 for all by 1978. The bill, approved Wednes-day 375 to 37, also would in-crease by 7 million to 56.5 mil-lion the number of workers cov-ered by the minimum wage laws. The bill is a modified version of one President Nixon vetoed last year. | WASHINGTON (AP) — The House passed a bill Wednesday raising the minimum wage to $2 an hour this year for most workers covered and to $2.30 for all by 1978. It also extends coverage to 7 million more per- sons, including household em-ployes. The bill, approved 375 to 37, is a modified version of the one President Nixon vetoed last year. |
| Greeley, Colo., Jan. 28. — Seven persons perished to-day in a fire that destroyed the home of Paul Martinez at Frederick, a small mining camp southwest of Gree-ley. : me The dead were Mrs. Martinez, her five children ranging in age from an infant a few months old to 12 years, and a man known as Newon who attempted to res- cue them. Martinez was at work at the time. ee: Newlon, operating pumps at mine near by, ran to the build-ing and shouted in an effort: to awaken the sleep-ing family. | FREDERICK, Colo. Jan. 28. (P) — Seven persons were burned to death in a fire to- day in a two-room shack 'near the Slopline mine. Mrs. — Paul Mar-tinez, her five chil-dren, ranging in age from 13 'months to 15 years, and a 'miner named Newlon lost their lives. Newton discovered the fire, 'broke into the shack through a window and was burned to death trying to rescue the family. he fire ig believed to have start- ed from an overheated stove. The father of the family was at work |
| PEPPERELL, Mass., Jan. 27 (7  -A 16-year old girl whose day dreams had brought her visions of being a fine titled lady of quality, — the sort of dreams all girls have has had those dreams come true. She had always believed herself to be humble Lucy Harriett Fagge, daughter of a humble choreman, 'who was so poor after her moth- ed died he couldn't care for her and sert her to her grandmother's home in Boston. There she had gone to grammer school and last fall had worked in a factory for a 'month. Yesterday, she came home to her dad. many years he had been Johnny Harry Lee Fagge, who worked at odd jobs about town, mowing lawns, mend-ing broken furniture and doing similar tasks. | PEPPERELL, Mass. Jan. 27. — (*) —- The dreams of wealth and po- Sition which all girls have came true today for 16-year-old Lucy Harriett Fagge. She had be-lieved herself to be the choreman, school and last fall had worked in @ factory. Her dad for many years had been Johnny Harry Lee Fagge. who, and the passing of the title and es- — tates to him. He asked for his Gaughter to share his new fortunes. — In his modest home here, Sir John told Lucy today of her ances-rive. Lucy isn't sure she wishes to go. "I have only been as far as the eighth grade in school." she said. "and ar-ent the daughters of the Utled men supposed to be very wise? |

Table A5: Examples of false positives from the bi-encoder model predictions. In all three cases, articles are about the same story and contain similar phrasing. Some examples have been truncated

### A.2.2 FALSE NEGATIVES

Among 50 randomly selected false negatives examined, in 16 cases at least one of the articles was very poorly OCR'ed (see for example, the first pair in table A6). In 15 cases, one of the articles was missing the first few paragraphs. This case can occur if an article is continued on an interior page. In four cases, one of the articles was heavily truncated (see for example, the first pair in table A6), and in another five cases, a significant proportion of the middle of an article was missing. Finally, in four cases, the pair was mislabeled, and in six cases there was no obvious cause of the error.

| | |
|---|---|
| — Kansas City, Harses, Jen, 26 — on the stow ¡n 2m ope¿ : ed the place todey 'where fire persous were crashed and 'burmed to death in a flamicg cz /Finne. — Fret Brke Laudemsn, apparently lexruggling with = failing moter back pene an efart to land als Trarelair [Sivanasence- carrier gt 2 time wb 'he could see the stingicg heecor and red mersers Qf Farriex airporr, hus goal, jess 2 mle azar. The airplane which few from. Wichita, ans late Mandar, Was behind schedule and dark Ness had closed ia when the irea- ; ble developed. Laudeman, losing ( altitude accor to witResse:. atung away from 4 buiding then - Went into = vertical bank te exash ; from sbout 150 feet... – i The cew of a se- icy enging near' [the scese sad the crit was envelop red 19 Fames hefore 10 Tel "The rotor jwes broLen in halt both sections ciz- ging inte the harm sround. The steel framesork, and utdereas age 'wae [oeisced and charred. Everyibing else, uichiding the bodies of che pilo¿ and Ins 2 four — Passengers, wes Ducted. | Kansas City, Kan. Jan. 28, WP — A black mark on (he snow 'in an open field here, marked the place tadey where five persons were crushed and buried to death in a Haming alrplone. Pilot Dyke Laudeman, apparentiy struggling with a failings motor had made an effort to land his Travel- nl six passenger carrier at a time when he could ser the aswhighng rhencgn and red markers of Fairfax, Alrport, nls goal, junta mile oway. The airplane which few fran Wichita, Kaun, lute Monday, was behlint seliedule and darkness had lelesed in when the traume de el. — oped. Lauder, losing altitude — Rcrordlyg to witnesses, swung away from a building- then went into P vertical bank lo crash trom about 150 feat, 'The crew of a switch engine near (he scene sold Lhe craft wis env 1- oped In flinnes before 16 fell. 'the motor was broken 1 hall, hoth sections digging into the hard bround, The steel framework, ane wnilercar- riage owas dwisted — and charred, Everythity else, In- elnd my the bodles af the pilot nnd his Tout Wssetgers, was burned, The dead were: Pilot Dyke Laudeman, Ka: City, Kan, Putsengers Miss Murgiret Dice, St. Joseya Mo. .R. Meltinnon, Chicago, recent |
| energy items were responsib] for about a fifth of last month' Increase in prices The Consumer Price Inde climbed last month to 1415 ¢ its 1987 average, mean- ing tha it cost ' consumers $141 90 ¢ buy the same amount of reta goods and services that $10 bought in 1987 While consumer prices cor tinued their sharp rise. rea spendable earnings of worker dropped another six- tenths one per centin February ani were down 45 per cent from , year ago This was the larges decline over a vear since th government began keeping that statistic in 1964 | WASHINGTON (AP) — The pace ..of inflation quick- ened in February with food and fuel prices pushing the cost of liv- ing up 13 per ce, the seearid biggest monthly jump since 4951, dhe government suid Lo- day. , The Labor Department said last month's rise sent con- sumer prices 10 per cent bigher than a year ago :and miarked th first time since 1948' that 'the United States experienced - double figure inllstion: -Was. the highest 12-month iuerease in the cost of living since: consumer prices rose by 10.2 per cent in the 12 months ending January 1948. Nearly half the February increase - was attributed a higher food prices with the price of beef raising 7.5 per eent, the sharpest jump since a 9.6 per cent increase in June 1947, Gasoline and other energy ilems were responsible for about a fifth of last month's increase m prices. The Consumer , Price Index climbed last month to 141.5 of its 1967 average, meaning that it cost consumers $141.50 to buy the same amount of retail gaorls and services thal $100 bonght in 1967. |
| WASHINGTON, Jan. 28. — Commenting on dis- patches from London announced | Washington, Jan, 28.— A PI Commenting on dis- patches from Tondon saying Great Britaln had an- nounced suspension of the cor-m of her two newest cr 3, Senator MeKeilar, Democrat, Tennessee, declared in the denate today: "They haven't cancelled anything, These ships never have been started. |

Table A6: Examples of false negatives from the bi-encoder model predictions. The first pair is an example of poor OCR. In the second, one of the articles is missing the beginning. In the third, one of the articles has been heavily truncated.

## A.3 EXAMPLES OF DUPLICATES IN C4

In this section, we provide randomly selected examples of predicted duplicates, organized by dataset, along with a brief discussion of the patterns that emerge.

| | |
|---|---|
| 1. Comedy Series: "Atlanta, " "Barry, " "black-ish, " "Curb Your Enthusiasm, " "GLOW, " "The Marvelous Mrs. Maisel, " "Silicon Valley, " Unbreakable Kimmy Schmidt "n2. Drama Series: "The Americans, " "The Crown, " "Game of Thrones, " "The Handmaid's Tale, " "Stranger Things, " "This Is Us, " "Westworld. "n3. Actor, Drama Series: Jason Bateman, "Ozark "; Sterling K. Brown, "This Is Us "; Ed Harris, "Westworld "; Matthew Rhys, "The Americans "; Milo Ventimiglia, "This Is Us "; Jeffrey Wright, "Westworld. "n4. Supporting Actor, Drama Series: Nikolaj Coster-Waldau, "Game of Thrones "; Peter Dinklage, "Game of Thrones "; Joseph Fiennes, "The Handmaid's Tale "; David Harbour, "Stranger Things "; Mandy Patinkin, "Homeland "; Matt Smith, "The Crown. "n5. Actress, Drama Series: Claire Foy, "The Crown "; Tatiana Maslany, "Orphan Black "; | 1. Comedy Series: "Atlanta, " "Barry, " "black-ish, " "Curb Your Enthusiasm, " "GLOW, " "The Marvelous Mrs. Maisel, " "Silicon Valley, " "Unbreakable Kimmy Schmidt "n6. Supporting Actress, Drama Series: Alexis Bledel, "The Handmaid's Tale "; Millie Bobby Brown, "Stranger Things "; Ann Dowd, "The Handmaid's Tale "; Lena Headey, "Game of Thrones "; Vanessa Kirby, "The Crown "; Thandie Newton, "Westworld "; Yvonne Strahovski, "The Handmaid's Tale. "n8. Supporting Actor, Comedy Series: Louie Anderson, "Baskets "; Alec Baldwin, "Saturday Night Live "; Tituss Burgess, "Unbreakable Kimmy Schmidt "; Brian Tyree Henry, "Atlanta "; Tony Shalhoub, "The Marvelous Mrs. Maisel "; Kenan Thompson, "Saturday Night Live "; Henry Winkler, "Barry. "n12. |
| ON the morning of Oct. 7, George Kimmerle of Mendham, N.J., decided he was fed up. He and his wife, Lynn, had spent the last two years trying to obtain zoning approval to renovate the small cottage in the borough of Stonington they had bought in 1996 as a weekend retreat. The cottage is next door to the 160-year-old Stonington Lighthouse, which once guided ships from Watch Hill, R.I., in the east to Fisher's Island, N.Y., in the west. On Oct. 7, Mr. Kimmerle learned that the Stonington Historical Society, which operates the lighthouse as a museum, was soliciting donations to fight his plan once again. The cottage is small, and the couple had at first wanted to build a 196-square-foot addition with large windows facing the lighthouse. | On the morning of Oct. 7, George Kimmerle of Mendham decided he was fed up. He and his wife, Lynn, had spent the last two years trying to obtain zoning approval to renovate a cottage they had bought in 1996 as a weekend retreat that sat next to the 160-year-old Stonington Lighthouse, with its dramatic ocean views of Watch Hill, R.I., and Fisher's Island, N.Y. On that day Mr. Kimmerle, an architect, had learned that the Stonington Historical Society, which operates the lighthouse as a museum, was soliciting donations to fight his plan. The cottage was small, and at first the couple wanted to build a 196-square-foot addition. After that plan was rejected by the Planning and Zoning Commission, they submitted one for 167 square feet. |
| Question papers of two subjects of the SSC (Class X) exam conducted by the Maharashtra State Board were found leaked at Bhiwandi in Thane district of Maharashtra on Wednesday, police said. A case was registered at Narpoli police station against an unidentified person in this connection. "As per the complaint filed by a state board official, examination of history and political science subjects was scheduled to take place on Wednesday," senior inspector M B Shinde of Narpoli police station said on Thursday. "For the exam that was to start at 11 am, students were expected to be in the exam hall by 10.15 am. However, outside an exam centre at Kalher in Bhiwandi, the board official found some girl students checking their mobile phones inside an autorickshaw," he added. | Question papers of two subjects of the SSC (Class X) exam conducted by the Maharashtra State Board were found leaked at Bhiwandi in Thane district of Maharashtra on Wednesday, police said. A case was registered at Narpoli police station against an unidentified person in this connection. "As per the complaint filed by a state board official, examination of history and political science subjects was scheduled to take place on Wednesday," senior inspector M B Shinde of Narpoli police station said on Thursday. "For the exam that was to start at 11 am, students were expected to be in the exam hall by 10.15 am. However, outside an exam centre at Kalher in Bhiwandi, the board official found some girl students checking their mobile phones inside an autorickshaw," he added. |

| | | |
|---|---|---|
| Star Wars characters will descend on Horsham town centre in an bid to raise charity funds. Gobsmack Comics will host its third annual Star Wars day, known as Force February, in Swan Walk on Saturday February 2. Drew Dewsall, the store's owner, is inviting residents to join a host of characters from a galaxy far far away in raising funds for the Springboard Project by dressing up as their favourite figures from the iconic film series. He said: "The aim is just to give the people of Horsham something different to do. | A host of Star Wars characters will descend on Horsham town centre next month. Drew Dewsall, the stores owner, is inviting residents to join a host of characters from a galaxy far far away in raising funds for the Springboard Project by dressing up as their favourite figures from the iconic film series. Visitors can expect a range of characters at the event. | Star Wars characters are heading to Horsham town centre to help raise charity funds this weekend. Gobsmack Comics is holding its third annual Star Wars day, known as Force February, in Swan Walk on Saturday (February 2) at 10am. Owner Drew Dewsall is inviting visitors to dress up as their favourite figures and give money to the Springboard Project. Drew, a lifelong Star Wars fan, said that visitors to the shop can expect to see Chewbacca, as well as stormtroopers, snowtroopers, jedi and Princess Leia, all in movie quality costumes. |
| US President Donald Trump directed his ire on Tuesday at the nation's major social media companies, claiming they're biased against Republicans and attacking them with the same gusto he uses for much of the rest of the media world. Trump's focus on social media began with a morning tweet and continued into a press conference with Brazilian President Jair Bolsonaro, where he reveled in the rightwing leader's use of Trump's trademark phrase "fake news". Trump's complaints come after he spent much of the weekend on Twitter, where he attacked a litany of targets including the late Sen. John McCain of Arizona, General Motors and "Saturday Night Live". | President Donald Trump speaks during a news conference with visiting Brazilian President Jair Bolsonaro, on the Rose Garden of the White House, March 19, 2019, in Washington. WASHINGTON — President Donald Trump directed his ire Tuesday at the nation's major social media companies, claiming they're biased against Republicans and attacking them with the same gusto he uses for much of the rest of the media world. Trump's focus on social media began with a morning tweet and continued into a press conference with Brazilian President Jair Bolsonaro, where he reveled in the rightwing leader's use of Trump's trademark phrase "fake news." | WASHINGTON - President Donald Trump directed his ire Tuesday at the nation's major social media companies, claiming they're biased against Republicans and attacking them with the same gusto he uses for much of the rest of the media world. Trump's comments came in response to a question about whether he could go along with legislation to make social media companies liable for their content on their platforms. Trump and some supporters have long accused Silicon Valley companies of being biased against them. Conservatives are complaining that those steps are disproportionally aimed at their side of the political spectrum. |

Table A7: Five randomly chosen clusters of duplicates predicted by the bi-encoder on RealNews. Three clusters had two articles and two clusters had three articles. Only one cluster contains a false positive (one of the articles about a Star Wars event). The final example shows a common phenomenon where articles have an additional first sentence, presumably the headline, but after that the article is a duplicate. Some articles have been truncated.

| | |
|---|---|
| Click here for the most recent update on the potential winter weather. The Piedmont Triad could see snow, ice and freezing rain over the weekend and it could continue into Monday. FOX8 MAX Weather Chief Meteorologist Van Denton said this is a "significant winter weather threat." Over the weekend, clouds will again thicken on Saturday as the next system pushes our way. Highs in the upper 30s to near 40. Light precipitation should start arriving over the southwestern counties during the afternoon and reach the Triad during the evening and overnight period. Early on, most precipitation should fall as rain and as the air cools, it will begin to mix with snow and sleet. Overnight Saturday, we expect rain and snow for areas south and east of the Triad and snow possibly mixed with rain for areas north and west. Lows in the upper 20s to near 30. For those that are getting rain, there will be freezing rain if these same spots slip below 32. Near the Triad, we may see our precipitation type change multiple times. Sunday we expect snow, mixed at times with rain and or sleet and/or freezing rain. Highs in the low to mid-30s. | Over the weekend, clouds will again thicken on Saturday as the next system pushes our way. Highs in the upper 30s to near 40. Light precipitation should start arriving over the southwestern counties during the afternoon and reach the Triad during the evening and overnight period. Early on, most precipitation should fall as rain and as the air cools, it will begin to mix with snow and sleet. Overnight Saturday, we expect rain and snow for areas south and east of the Triad and snow possibly mixed with rain for areas north and west. Lows in the upper 20s to near 30. For those that are getting rain, there will be freezing rain if these same spots slip below 32. Near the Triad, we may see our precipitation type change multiple times. Sunday we expect snow, mixed at times with rain and or sleet and or freezing rain. Highs in the low to mid-30s. Monday there is still a chance for some snow showers, with lows in the upper 20s with highs in the mid-30s. Dry weather returns for Tuesday through Thursday with rain returning next Friday. Highs Tuesday through Thursday from near 40 back to the mid-40s. We head back to 50 next Friday. |
| Israeli forces use stun grenades and worshippers throw stones during clashes at the mosque in occupied East Jerusalem. Israeli forces raided the mosque compound and fired stun grenades on Friday, while dozens of worshipers threw stones and chanted: "We sacrifice our blood and souls for you Aqsa". The imam of the mosque, Mohamed Hussian, condemned the violence at one of Islam's holiest sites - known to Jews as Temple Mount. "It is a clear violation which is rejected by all the religions and the international laws, " he said. "It is a violation against al-Aqsa mosque and the Israeli authorities are responsible, because they order their soldiers to raid the mosque violently, they are responsible for all what is happening in Al-Aqsa mosque. " "Israeli police units responded by using stun grenades and entering inside the Temple Mount area, immediately we made sure that we dispersed all the rioters. " | Clashes have erupted between Israeli forces and worshippers at the al-Aqsa mosque compound in occupied East Jerusalem after noon prayers. Israeli forces raided the mosque compound and fired stun grenades on Friday, while dozens of worshipers threw stones and chanted: "We sacrifice our blood and souls for you Aqsa". One man was wounded and treated inside the mosque compound. The imam of the mosque, Mohamed Hussian, condemned the violence at one of Islam's holiest sites - known to Jews as Temple Mount. "It is a clear violation which is rejected by all the religions and the international laws," he said. Israeli police spokesperson Micky Rosenfeld said police had responded after stones were thrown at them. Rosenfeld said police had arrested seven people during the two-hour operation. |
| The addition of Secure Sockets Layer technology means users will be able to send e-mail that will stay private until it reaches their Internet service provider. After that, the e-mail goes decrypted over the Internet cloud - a space filled with so many billions of bytes, it's very difficult to spy on a single message. Eudora's SSL system then re-encrypts the message at the recipient's ISP, which gives it the same protection at the receiving end as it had when it was sent. "The question is: Where is the attack going to take place? " said Kawika Daguio, president of OS Crypto, a security consulting firm. "Most of the time, it's someone watching one of the two parties and not [trying to eavesdrop on] everything. " Unfortunately, sophisticated attackers may have other ways of reading people's e-mail, said David Crocker, director at the Internet Mail Consortium. And while SSL requires no effort by the end user, the fact that mainstream programs such as Eudora are only now adopting it suggests that the state of e-mail security is far behind that of the rest of the Net, he said. "Security technology in e-mail is very, very poor, " Crocker said. | Qualcomm, maker of the Eudora e-mail program, announced it will add seamless privacy capabilities to the world's most popular retail e-mail client. Qualcomm, maker of the Eudora e-mail program, announced it will add seamless privacy capabilities to the worlds most popular retail e-mail client. The addition of Secure Sockets Layer technology means users will be able to send e-mail that will stay private until it reaches their Internet service provider. After that, the e-mail goes decrypted over the Internet cloud — a space filled with so many billions of bytes, its very difficult to spy on a single message. Eudoras SSL system then re-encrypts the message at the recipients ISP, which gives it the same protection at the receiving end as it had when it was sent. "The question is: Where is the attack going to take place? " said Kawika Daguio, president of OS Crypto, a security consulting firm. "Most of the time, its someone watching one of the two parties and not [trying to eavesdrop on] everything. " Unfortunately, sophisticated attackers may have other ways of reading peoples e-mail, said David Crocker, director at the Internet Mail Consortium. |

| | |
|---|---|
| Reuters An anti-U.S. mural in Tehran, October 2017. Trump apparently wishes not merely to contain Iran's power but to roll back its regional presence, confining its influence to its borders, disarming it, and, by implication, changing its regime, given that these are constraints that Iran's government could not tolerate for profound strategic and ideological reasons. Doing so would take a massive effort and likely entail another American war in the Middle East—one that the president is not committed to fighting and would not have the popular support to pursue. Rather than a coherent strategy, Trump's aggressive behavior reflects a strange and unhealthy obsession with Iran unwarranted by the actual threat it poses to the interests of the United States and its allies. The risk now is that the United States could drift into a war with Iran in a fog of bombastic threats and jolting policy reversals even if there were no underlying interest in hostilities. But although Trump's rhetoric is dangerous, his administration's inordinate antagonism is rooted in a deeper inability, going all the way back to 1979, of the United States to find a way forward with Iran. It is time for Washington to do so before it is too late. | STEVEN SIMON is a Professor at Amherst College and served on the National Security Council in the Clinton and Obama administrations. JONATHAN STEVENSON is Senior Fellow for U.S. Defense and Editor of Strategic Comments at the International Institute for Strategic Studies. He served as Director for Political-Military Affairs for the Middle East and North Africa on the U.S. National Security Council staff from 2011 to 2013. Trump apparently wishes not merely to contain Iran's power but to roll back its regional presence, confining its influence to its borders, disarming it, and, by implication, changing its regime , given that these are constraints that Iran's government could not tolerate for profound strategic and ideological reasons. Doing so would take a massive effort and likely entail another American war in the Middle East—one that the president is not committed to fighting and would not have the popular support to pursue. Rather than a coherent strategy, Trump's aggressive behavior reflects a strange and unhealthy obsession with Iran unwarranted by the actual threat it poses to the interests of the United States and its allies. |
| According to the District of Columbia 's Metropolitan Police Department, the nation's capital reported 135 homicides last year. One of those homicides, the killing of Democratic National Committee staffer Seth Rich on July 10, 2016, continues to make news ten months later. Who killed Seth Rich, and why? We may never know for sure. On the other hand, a significant piece of the puzzle may have just fallen into place. Fox News, citing a federal investigator as a source, reports that Rich may well — as long rumored — have been the source of DNC emails published by WikiLeaks, less than two weeks after he was shot twice in the back during a robbery in which, curiously, nothing was apparently taken from him. That email release, which revealed an internal DNC conspiracy to ensure the nomination of Hillary Clinton for president at the expense of her opponent, Bernie Sanders, wounded Clinton's campaign and cost US Representative Debbie Wasserman Schultz her position as DNC chair. | Fox News, citing a federal investigator as source, reports that Rich may well - as long rumored - have been the source of DNC emails published by WikiLeaks, less than two weeks after he was shot twice in the back during a robbery in which, curiously, nothing was apparently taken from him. That email release, which revealed an internal DNC conspiracy to ensure the nomination of Hillary Clinton for president at the expense of her opponent, Bernie Sanders, wounded Clinton's campaign and cost US Representative Debbie Wasserman-Schultz her position as DNC chair. WikiLeaks founder/director Julian Assange, in line with the organization's policy against outing sources, has resolutely declined to confirm or deny Rich as the DNC leaker. On the other hand, WikiLeaks did put up reward money for information leading to the arrest and conviction of his killer or killers - and retweeted, without comment, the Fox News story referenced above. |

Table A8: Five randomly chosen clusters of duplicates predicted by LSH among RealNews. All five clusters happened to have two articles (the modal value). Two pairs appear to be actual duplicates, while the other three contain similar topics, but are not duplicated. Some articles have been truncated.

| | |
|---|---|
| 2014-10-01 Assigned to HAND HELD PRODUCTS, INC. reassignment HAND HELD PRODUCTS, INC. ASSIGNMENT OF ASSIGNORS INTEREST (SEE DOCUMENT FOR DETAILS). Assignors: QU, HUYU, WANG, YNJIUN P. A portable encoded information reading (EIR) terminal for incorporation in a data collection system having a host computer, a plurality of peer EIR terminals, and a plurality of interconnected networks including one or more wireless networks, can comprise a central processing unit (CPU), a memory, an encoded information reading (EIR) device, and at least one wireless communication interface. The EIR terminal can be associated with a home network and have a home address belonging to the address range associated with the home network. The EIR terminal can participate in one or more communication sessions and exchange messages, at least one of which can include decoded message data corresponding to an encoded message, with the host computer. | 2010-07-02 Assigned to HAND HELD PRODUCTS, INC. reassignment HAND HELD PRODUCTS, INC. ASSIGNMENT OF ASSIGNORS INTEREST (SEE DOCUMENT FOR DETAILS). Assignors: QU, HUYU, WANG, YNJIUN P., HAVENS, WILLIAM H. A portable data terminal (PDT) adapted to participate in a wireless mesh network including a plurality of peer PDTs can comprise: a PDT module including an encoded information reading (EIR) device, and a mesh point (MP) module communicatively coupled to the PDT module. The MP module can include a microcontroller and at least one wireless communication interface and can be configured to perform IEEE 802.11-conformant wireless station services including authentication, de-authentication, privacy, and MAC service data unit delivery, and IEEE 802.11-conformant wireless distribution system services including association, disassociation, distribution, integration, and re-association |
| An instrument useful in performing lumbar arthrodesis with a minimal approach which spares the lumbar muscles from surgical disruption and includes one of two retractor designs having blades angled approximately $90\breve{0}b0$ with respect to each respective retractor handle. One blade is bent at an end portion thereof in a direction away from the handle portion. The other blade has first and second blade faces, with the second face having at least two toothed structures located thereon. This application is a continuation of U.S. patent application Ser. No. 09/969,138 filed on Oct. 1, 2001 and entitled "METHOD AND DEVICE FOR RETRACTOR FOR MICROSURGICAL INTERMUSCULAR LUMBAR ARTHRODESIS", now U.S. Pat. No. 6,692,434 | This application is a continuation of U.S. patent application Ser. No. 09/969,138 filed on Oct. 1, 2001 and entitled "METHOD AND DEVICE FOR RETRACTOR FOR MICROSURGICAL INTERMUSCULAR LUMBAR ARTHRODESIS", which claimed priority from U.S. Provisional Patent Application No. 60/236,584 filed on Sep. 29, 2000. The entire disclosures of these applications are considered to be part of the disclosure of the present application and are hereby incorporated by reference in their entirety. The present invention is directed to a method and device for performing an instrumented lumbar interbody fusion utilizing a minimally invasive approach. |
| 2007-02-13 Assigned to 3M INNOVATIVE PROPERTIES COMPANY reassignment 3M INNOVATIVE PROPERTIES COMPANY ASSIGNMENT OF ASSIGNORS INTEREST (SEE DOCUMENT FOR DETAILS). Assignors: VOS, MARTIN J. An untethered stylus, configured to cooperate with a location sensor, includes a coil resonant circuit configured to develop an arbitrary AC voltage in response to a varying magnetic field produced by the location sensor. The coil resonant circuit includes a first capacitor and an inductive coil. A power converter includes a switch circuit having an output coupled to a second capacitor, an input coupled to the coil resonant circuit, and a threshold voltage. The switch circuit facilitates charging of the second capacitor in response to the arbitrary AC voltage and discontinuance of second capacitor charging in response to a voltage across the first capacitor reaching the threshold voltage so as to prevent diversion of a discharging current when the arbitrary AC voltage exceeds the threshold voltage. A stable DC voltage is provided at the output of the switch circuit. The power converter is preferably devoid of a Zener diode. | 2007-03-19 Assigned to 3M INNOVATIVE PROPERTIES COMPANY reassignment 3M INNOVATIVE PROPERTIES COMPANY ASSIGNMENT OF ASSIGNORS INTEREST (SEE DOCUMENT FOR DETAILS). Assignors: LIBBEY, ALBERT H., VOS, MARTIN J. An untethered stylus is configured to cooperate with a location sensing device that generates a continuously varying magnetic drive signal that powers the stylus. The stylus includes a housing having a tip, a shield, and an antenna arrangement coupled between the tip and shield. A resonant circuit of the antenna arrangement is tuned to a frequency of the magnetic drive signal. An oscillator circuit, coupled to and powered by the antenna arrangement, is configured to oscillate at a frequency corresponding to data to be communicated from the stylus, and to amplitude modulate a voltage developed at the stylus tip at the oscillator circuit frequency. Repetitive current draw from the antenna arrangement to power the oscillator circuit repetitively reduces a voltage at the tip, such that the tip voltage is amplitude modulated at the oscillator circuit frequency. |

| | | |
|---|---|---|
| 2002-03-13 Assigned to TESSERA, INC. reassignment TESSERA, INC. ASSIGNMENT OF ASSIGNORS INTEREST (SEE DOCUMENT FOR DETAILS). Assignors: PICK-ETT, CHRISTOPHER M., SMITH, JOHN W. A microelectronic assembly is made by bonding the tip ends of leads on a first element to bonding contacts on a second element. The tip ends of the leads are releasably connected to the first element, so that the leads are held in place during the bonding process. After bonding, the first and second elements are heated or cooled to cause differential thermal expansion, which breaks at least some of the releasable attachments of the tip ends, leaving the leads free to flex. | 2009-05-06 Assigned to TESSERA, INC. reassignment TESSERA, INC. ASSIGNMENT OF ASSIGNORS INTEREST (SEE DOCUMENT FOR DETAILS). Assignors: FJELSTAD, JOSEPH C. A microelectronic assembly is provided which can include an element including a first dielectric layer and a second dielectric layer overlying the first dielectric layer, the second dielectric layer having an exposed surface defining an exposed major surface of the element. A plurality of substantially rigid metal posts can project beyond the exposed surface, the metal posts having ends remote from the exposed surface. The microelectronic assembly can include a microelectronic device which has bond pads and overlies the element. | 1996-12-27 Assigned to TESSERA, INC. reassignment TESSERA, INC. ASSIGNMENT OF ASSIGNORS INTEREST (SEE DOCUMENT FOR DETAILS). Assignors: GILLEO, KENNETH B. A microelectronic element assembly such as a semiconductor chip assembly uses a connection component incorporating a dielectric sheet with electrically conductive elements therein. Each electrically conductive element may include a flexible shell. The flexible shells can be formed to assure reliable engagement with mating contact pads. The present application claims benefit of U.S. Provisional Application No. 60/001,669, filed Jul. 31, 1995. |
| The present invention relates in certain embodiments to methods, systems, and devices for treating vertebral compression fractures. In one embodiment, a bone cement injector is advanced into bone and injects a bone cement flow through the injector and a vapor flow from at least one vapor outlet in the injector. In another embodiment, the bone treatment system comprises an elongated member having a flow passageway, a bone fill material source, and a vapor source coupleable to the flow passageway. | Systems and methods for treating vertebral compression fractures are provided. In one embodiment, a bone cement injector system can include a first handle component that is detachably coupled to a second sleeve component having a distal end configured for positioning in bone, and a flow channel extending through the first and second components. The system can include a thermal energy emitter. The flow channel can have a flow channel surface with a material that that limits cement flow turbulence. | The present invention relates in certain embodiments to systems for treating vertebral compression fractures. In one embodiment, a trocar with a flexible tip is provided to create a curved path in cancellous bone. An injector can be introduced into the vertebra in communication with the curved path for delivery of bone fill material into the curved path. Optionally, thermal energy can be applied to the bone fill material prior to injection into the curved path in cancellous bone to alter a property (e.g., viscosity) of the bone fill material. |

Table A9: Five randomly chosen clusters of duplicates predicted by the bi-encoder among Google Patents. Three clusters have two patents while two clusters have two patents. The first four are extremely close and appear to be slightly modified versions of the same or similar patents, including several segments of duplicated text. The final cluster is an exact duplicate. All patents have been truncated.

| | |
|---|---|
| An optical amplifier of a wavelength-division multiplexing transmission system that includes a pre-stage optical amplifying unit and a post stage optical amplifying unit. The pre-stage optical amplifying unit has a rare-earth element doped optical fiber and a pump light source for inputting a pump light to the rare-earth element doped optical fiber. The post-stage optical amplifying unit has a second rare-earth element doped optical fiber and a pump light source for inputting a pump light to the second rare-earth element doped optical fiber. | A wavelength-division multiplexing optical communication system for wavelength-division multiplexing a plurality of optical signals having different wavelengths and transmitting a wavelength-division multiplexed signal via an optical fiber transmission line. The wavelength-division multiplexing system includes a multiplexing unit that wavelength-multiplexes a plurality of optical signals having different wavelengths. A storing unit stores information regarding an addition or subtraction of an optical signal to be wavelength-multiplexed |
| Methods, apparatuses and systems directed to sponsored story generation from an organic activity stream in a social networking site. A user wishing to promote an entry from an organic activity stream may, using a sponsor user interface, specify the types of stories to promote to a portion of the home page displayed to a member of a social network. This application is a continuation-in-part of application Ser. No. 12/193,702, filed Aug. 18, 2008, which claims priority to U.S. Provisional Application Ser. No. 60/985,631, filed Nov. 5, 2007. | A method includes monitoring an activity stream to identify actions that match stored sponsored story specifications, for providing one or more sponsored stories to a viewing user. The sponsored story specifications include a visual specification for the sponsored story, and matched sponsored stories are ranked for a viewing user. Users can set privacy preferences related to sponsored stories. The ranking and privacy settings contribute to which sponsored stories are provided for display to the viewing user. This application is a continuation of U.S. patent application Ser. No. 13/488,596, filed Jun. 5, 2012, |
| 1999-12-23 Assigned to 3COM CORPORATION reassignment 3COM CORPORATION ASSIGNMENT OF ASSIGNORS INTEREST (SEE DOCUMENT FOR DETAILS). Assignors: BELKIND, RONNEN, GRABIEC, JACEK A., SIDHU, IKHLAQ S., SCHUSTER, GUIDO M. 2013-05-23 Assigned to QUALCOMM INCORPORATED reassignment QUALCOMM INCORPORATED ASSIGNMENT OF ASSIGNORS INTEREST (SEE DOCUMENT FOR DETAILS). | 2000-08-23 Assigned to 3COM CORPORATION reassignment 3COM CORPORATION ASSIGNMENT OF ASSIGNORS INTEREST (SEE DOCUMENT FOR DETAILS). Assignors: DEAN, FREDERICK D., GRABIEC, JACEK A., MAHLER, JERRY J., SCHUSTER, GUIDO M., SIDHU, IKHLAQ S. A system and method for communicating screen display images on a computer to another computer using a telephony network. A screen share button on a data network telephone initiates a screen shot request to a first computer associated with the data network telephone. |
| A catheter assembly employs an outer catheter with a pre-formed distal end and an open lumen. An inner catheter having an open lumen and a pre-formed distal end is movably disposed within the outer catheter. Relative rotation and extension of the inner and outer catheters provides the distal end of the catheter assembly with an adjustable range of two- and three-dimensional shapes. The inner catheter can include sections of varying stiffness, such that extension of the inner catheter within the outer catheter modifies the shape of the outer catheter's pre-formed distal end. | A method of delivering a payload to a destination vessel branching from a coronary sinus of a patient's heart. The method comprises inserting a catheter assembly into a right atrium of the patient's heart. The catheter assembly comprises an outer catheter and an inner catheter movably disposed within the open lumen of the outer catheter. The inner and outer catheters are operable to be rotated and translated relative to one another such that the distal end of the outer catheter can assume a selectable plurality of shapes appropriate for accessing the coronary sinus. |
| The present invention relates to high-throughput systems for analyzing samples by both liquid chromatography and mass spectrometry. Mass spectrometry (MS) is an important analysis technique in many industrial and academic fields. MS exploits the behavior of the gas-phase ions (i.e., gaseous molecules with a non-zero charge) of a molecule in response to applied electric and magnetic fields in order to deduce the composition of the molecule. The ionization process breaks a molecule into its components, the mass of each of which is then determined by analyzing the trajectory of the components as they travel through the mass spectrometer. | 2003-08-18 Assigned to NANOSTREAM, INC. reassignment NANOSTREAM, INC. ASSIGNMENT OF ASSIGNORS INTEREST (SEE DOCUMENT FOR DETAILS). Assignors: COVINGTON, JOSEPH F., GREGORI, MATTHEW M., HOBBS, STEVEN E. Systems and methods for collecting the output of multiple simultaneously operated chromatography columns and providing the outputs to a single mass spectrometer are provided. Such systems utilize predetermined lengths of microfluidic tubing that act as storage buffers for the substantially all of the output of each column. |

Table A10: Five randomly chosen clusters of duplicates predicted by LSH among Google Patents. All five clusters happened to have two patents. The patents share some similarities, and the third example contains duplicated boilerplate language, but none of them are obvious duplicates. All patents have been truncated

| | |
|---|---|
| Fossil fuel – A fossil fuel is a fuel formed by natural processes, such as anaerobic decomposition of buried dead organisms, containing energy originating in ancient photosynthesis. The age of the organisms and their resulting fossil fuels is typically millions of years, and sometimes exceeds 650 million years. | Fossil fuels are fuels formed by natural processes such as anaerobic decomposition of buried dead organisms. The age of the organisms and their resulting fossil fuels is typically millions of years, and sometimes exceeds 650 million years. Fossil fuels contain high percentages of carbon and include coal, petroleum, and natural gas. |
| Kennedy Space Center – The John F. Kennedy Space Center (KSC, originally known as the NASA Launch Operations Center) is one of ten National Aeronautics and Space Administration field centers. Since December 1968, Kennedy Space Center has been NASA's primary launch center of human spaceflight. Launch operations for the Apollo, Skylab and Space Shuttle programs were carried out from Kennedy Space Center Launch Complex 39 and managed by KSC. | The John F. Kennedy Space Center is one of ten National Aeronautics and Space Administration field centers. Since December 1968, Kennedy Space Center has been NASA's primary launch center of human spaceflight. Launch operations for the Apollo, Skylab and Space Shuttle programs were carried out from Kennedy Space Center Launch Complex 39 and managed by KSC. |

Table A11: Examples of duplicates predicted by bi-encoder from RealNews in the BoolQ development set.

| | |
|---|---|
| Washington (CNN) Donald Trump said Sunday that Russian President Vladimir Putin won't make a military move into Ukraine – even though Putin already has done just that, seizing the country's Crimean Peninsula. "He's not going into Ukraine, OK, just so you understand. He's not going to go into Ukraine, all right? You can mark it down. You can put it down. You can take it anywhere you want," Trump said in an interview on Sunday with ABC's George Stephanopoulos on "This Week. " | Washington(CNN) Donald Trump said Sunday that Russian President Vladimir Putin won't make a military move into Ukraine – even though Putin already has done just that, seizing the country's Crimean Peninsula. "He's not going into Ukraine, OK, just so you understand. He's not going to go into Ukraine, all right? You can mark it down. You can put it down. You can take it anywhere you want, " Trump said in an interview on Sunday with ABC's George Stephanopoulos on "This Week. " |
| (CNN) After a slow start in October, flu season in the United States is gaining speed, particularly in the South. Flu activity, which has been increasing since the start of November, is now higher than usual for this time of year, according to a report published Thursday by the Centers for Disease Control and Prevention. Flu is a contagious, viral illness that causes mild to severe symptoms that, in rare cases, can lead to death. | After a slow start in October, flu season in the United States is gaining speed, particularly in the South. Flu activity, which has been increasing since the start of November, is now higher than usual for this time of year, according to a report published Thursday by the Centers for Disease Control and Prevention. Flu is a contagious, viral illness that causes mild to severe symptoms that, in rare cases, can lead to death. |

Table A12: Examples of duplicates predicted by bi-encoder from RealNews in the ReCoRD development set.

| | |
|---|---|
| Plans are being drawn up to build a £3.3m working replica of the boat that took Charles Darwin around the world at Milford Haven in Pembrokeshire. Fundraising for the project, which would mark the 200th anniversary of Darwin's birth in 2009, is under way. The aim is to built a seaworthy vessel identical to the HMS Beagle on the outside, but with a modern interior. | Plans are being drawn up to build a £33.3m working replica of the boat that took Charles Darwin around the world at Milford Haven in Pembrokeshire. Fundraising for the project, which would mark the 200th anniversary of Darwin's birth in 2009, is under way. The aim is to built a seaworthy vessel identical to the HMS Beagle on the outside, but with a modern interior. |

Table A13: Examples of duplicates predicted by bi-encoder from RealNews in the RTE development set.

## A.4   LARGE CLUSTERS IN C4

When de-duplicating C4, we removed any clusters from the results that contained over 1000 texts, for RealNews, or over 300 texts, for Google Patents. This leads to the removal of 27 clusters for the bi-encoder over RealNews and 7 clusters for the bi-encoder over Google Patents. No clusters are removed under LSH. These large clusters are not errors, as such, but tend to be clusters of texts that all follow some template. It seems possible that these would also cause problems for training large language models. However, there are also some cases where it is hoped that LLMs will learn to generate texts following such templates. This appendix gives some examples of these templates.

| |
|---|
| 27% of this provider's 380 patients filled at least one prescription for an antibiotic drug, compared to an average of 24%. 18% of this provider's 380 patients filled at least one prescription for an opioid, compared to an average of 14%. |
| 32 of this provider's 74 patients who are 65 and older filled at least one prescription for an antipsychotic drug. 0% of this provider's 146 patients filled at least one prescription for an antibiotic drug, compared to an average of 1% |
| 2% of this provider's 532 patients who are 65 and older filled at least one prescription for an antipsychotic drug, compared to an average of 3%. 44% of this provider's 618 patients filled at least one prescription for an antibiotic drug, compared to an average of 29% |

| |
|---|
| Select up to 3 trims below to compare some key specs and options for the 2010 Mercury Milan Hybrid. For full details such as dimensions, cargo capacity, suspension, colors, and brakes, click on a specific Milan Hybrid trim. |
| Select up to 3 trims below to compare some key specs and options for the 2010 BMW X6. For full details such as dimensions, cargo capacity, suspension, colors, and brakes, click on a specific X6 trim. |
| Select up to 3 trims below to compare some key specs and options for the 2012 Buick Enclave. For full details such as dimensions, cargo capacity, suspension, colors, and brakes, click on a specific Enclave trim. |

| |
|---|
| 64, of Las Vegas, died in Las Vegas on November 25, 2018. She was born in Honolulu, Hawaii. Visitation: 10 a.m.; Services: 11 a.m. on Saturday, February 9, 2019 at Blessed Sacrament Church. Inurnment: 12:45 p.m. at Nuuanu Memorial Park. |
| 86, of Aiea, died in Honolulu on January 24, 2019. He was born in Pulehu, Maui. Visitation: 10:30 AM; Services: 11:30 AM on Monday, February 11, 2019 at Borthwick Mortuary Vineyard Chapel. Inurnment: 2 PM at National Memorial Cemetery of the Pacific. |
| 87, of Hilo, Hawaii, died in Hilo on December 18, 2017. She was born in Hilo. Visitation: 6:00 p.m. Wake Service: 7:00 p.m. on Wednesday, December 27, 2017 at Dodo Mortuary Chapel in Hilo. Mass: 10:30 a.m. on Thursday, December 28th at Saint Joseph Catholic Church in Hilo. |

Table A14: Examples of a large clusters from RealNews. There are over 1000 articles that fit each of these templates.

2003-09-10 First worldwide family litigation filed litigation Critical `https://patents.darts-ip.com/?family=20401150&utm_source=google_patent&utm_medium=platform_link&utm_campaign=public_patent_search&patent=US20060293642(A1)` "Global patent litigation dataset" by Darts-ip is licensed under a Creative Commons Attribution 4.0 International License. The present invention relates to wetting apparatus for wetting or hydrophilic urinary catheters comprising a wetting receptacle which defines a wetting fluid receiving area which is adapted to receive a hydrophilic urinary catheter

2005-04-21 First worldwide family litigation filed litigation Critical `https://patents.darts-ip.com/?family=21928393&utm_source=google_patent&utm_medium=platform_link&utm_campaign=public_patent_search&patent=EP0719564(B1)` "Global patent litigation dataset" by Darts-ip is licensed under a Creative Commons Attribution 4.0 International License. This invention relates generally to medical appliances; and more particularly to a device for inserting a cannula – such as an intravenous cannula – into a patient's body. As is well known, there are myriad very important medical uses

2003-01-06 First worldwide family litigation filed litigation Critical `https://patents.darts-ip.com/?family=40090178&utm_source=google_patent&utm_medium=platform_link&utm_campaign=public_patent_search&patent=US6554611(B2)` "Global patent litigation dataset" by Darts-ip is licensed under a Creative Commons Attribution 4.0 International License. A system for repositioning teeth comprises a plurality of individual appliances. The appliances are configured to be placed successively on the patient's teeth and to incrementally reposition the teeth from an initial tooth arrangement,

Table A15: Examples of a large cluster from Google Patents. There are over 300 articles that follow this template.

## A.5 IMPACT OF TEST SET LEAKAGE FROM SUPERGLUE IN REALNEWS

Table 5 reports test set leakage from RealNews to SuperGlue.

From the test sets that we considered, we found that ReCoRD, which is only part of SuperGLUE that comprises entirely of news articles, contains a particularly high number of texts with near duplicates in RealNews.

ReCoRD is a question-answering task, composed of a passage from a news article and a query with a missing entity, marked @placeholder. The task is for the model to determine which word @placeholder represents. In many cases, where we found a duplicate in RealNews, it was a longer version of the same passage, which included the query sentence, with the correct word in place of @placeholder

For example, this passage and query pair appears in ReCoRD:

Passage: "Minneapolis (CNN) The mayor of Minneapolis said she wants to hear from the officer who fatally shot Justine Ruszczyk. But so far, officer Mohamed Noor has exercised his constitutional right to not speak to state investigators, the Minnesota Bureau of Criminal Apprehension said Tuesday. And, it's not clear if or when he will. "He has a story to tell that no one else can tell," Mayor Betsy Hodges said in a news conference Tuesday. "We can't compel him by law, but I wish that he would make that statement." The news conference capped a day of developments in a case that's raising questions about police training, use of force and body camera policies. The shooting has led newscasts in Australia, where Ruszczyk is originally from. @highlight A county attorney, not a grand jury, will decide whether either of the officers will be charged. @highlight Family shown documents from case ahead of public release, council member says"

Query: "Those documents were shared with @placeholder's family Tuesday night, she said."

The following near duplicate appears in RealNews:

"MINNEAPOLIS - The mayor of Minneapolis said she wants to hear from the officer who fatally shot Justine Ruszczyk. But so far, officer Mohamed Noor has exercised his constitutional right to not speak to state investigators, the Minnesota Bureau of Criminal Apprehension said Tuesday. And, it's not clear if or when he will. The news conference capped a day of developments in a case that's raising questions about police training, use of force and body camera policies. The shooting has led

newscasts in Australia, where Ruszczyk is originally from. The department's BCA investigation is expected to last two to four months, said Chuck Laszewski, spokesman for the Hennepin County attorney's office. Once that happens, county attorney Mike Freeman - not a grand jury - will decide whether either of the two officers involved should be charged in Ruszczyk's death. Meanwhile, frustration over the lack of information grows. City Council member Linea Palmisano said some documents from the case would be released online Wednesday morning. Those documents were shared with Ruszczyk's family Tuesday night, she said. [Truncated]"

Therefore in these cases of duplicates, when a model has been trained on C4 (and therefore Real-News) it has been trained on data containing the correct answers to many of the texts in the ReCoRD test set.

To evaluate the practical impact of this, we evaluated 100 examples from ReCoRD, zero-shot, with GPT-3, which was trained on Common Crawl (including RealNews). 50 of these are examples that had near duplicates in RealNews and 50 were examples without near duplicates. We prompted GPT-3 with the passage, the query and "Using the information above, what word or words should be part of the final sentence, instead of @placeholder?". For the examples with near duplicates in RealNews accuracy was 84%, compared to 70% on the examples without a near duplicate in the training set. This small evaluation provides suggestive evidence that test set leakage artificially inflates model performance.

