# OpenReview forum: "Noise-Robust De-Duplication at Scale"
_ICLR.cc/2023/Conference — ICLR 2023 poster_

### Official Review · Reviewer_9x67 · 2022-10-22

**Confidence:** 3
**Correctness:** 3
**Technical Novelty And Significance:** 3
**Empirical Novelty And Significance:** 2
**Recommendation:** 6

**Clarity, Quality, Novelty And Reproducibility:**

Hi, I'm currently on a medical leave and won't be able to perform ICRL review duties. sorry for the late notice.

**Strength And Weaknesses:**

Hi, I'm currently on a medical leave and won't be able to perform ICRL review duties. sorry for the late notice.

**Summary Of The Paper:**

Hi, I'm currently on a medical leave and won't be able to perform ICRL review duties. sorry for the late notice.

**Summary Of The Review:**

Hi, I'm currently on a medical leave and won't be able to perform ICRL review duties. sorry for the late notice.

---

### Official Review · Reviewer_p1KW · 2022-10-24

**Confidence:** 4
**Correctness:** 3
**Technical Novelty And Significance:** 1
**Empirical Novelty And Significance:** 3
**Recommendation:** 8

**Clarity, Quality, Novelty And Reproducibility:**

Generally the paper is clear (see a few textual notes below) and I feel like the dataset and experiments are novel within their niche, although its niche is fairly small (dedupliicating historical texts).  The paper does not propose novel methods.

Sec 3.2 “days of content 1930” -> “days of content in 1930”

The hand-removed articles that had incorrect bounding boxes—what fraction was this?

The sentence “duplicates are defined…” seems to be missing something about the articles being about the same story, not just from the same service.

“Congruence labeling was used”--I don’t know what this means, did you measure interannotator agreement?  I don't see this in the paper.


**Strength And Weaknesses:**

Strengths

The paper is well-written and introduces a resource and a baseline evaluation that will be of interest to a subset of the community.  I felt the experiments were generally well-done, although I had some questions and suggestions as detailed below.

Weaknesses

This paper is really more about OCRed text than the title and abstract let on---the failure of the standard n-gram-based methods which work well on born-digital content fail here.  Clarifying this in the title and abstract might help ensure the paper reaches the right audience.

Similarly, I was not sure that methods for correcting or compensating for the OCRed text were adequately explored here.  The paper does not include concrete examples of what the noisy n-grams look like (this should be added to support the statistics about n-gram noise early in the paper), but it would seem plausible that simple normalization techniques such as removing whitespace and using character n-grams, or mapping each word to a fixed dictionary, might mitigate some of the noise.  Techniques like this are left unexplored in the paper.

A few additional questions.  How does the accuracy of the re-ranker change as the bi-encoder threshold for re-ranking varies?  Further, the statement about the complexity of the cross-encoder (that it would require 10^14 cross-encodings) seems way too pessimistic given that you could use some kind of temporal constraint to decide which comparisons are worth trying (as I understood it---you have accurate metadata like date and news service).  As a separate point, I was also unclear on whether some kind of temporal constraint (not allowing clustering across large time spans) would have helped in the experiments reported in Table 2.  My guess is not, given the types of errors you describe that that evaluation only considers four distinct days, but it might be helpful to discuss this.




**Summary Of The Paper:**

This paper introduces a new data set for deduplication historical news articles, and evaluates several baseline algorithms on it, finding that neural methods perform very well compared to n-gram-based baselines.

**Summary Of The Review:**

This paper is fairly narrowly-focused and does not present novel methods, and the dataset while helpful is fairly small and limited in scope (and at least one critical detail for a dataset paper, annotator agreement, is not reported).  However, I feel like the results are valid (even if some of the n-gram methods are underexplored) and would be of interest to some portion of the community.

---

> ### Author Response · Authors · 2022-11-18
> **Response to reviewer p1KW (part 1)**
>
> > “This paper is really more about OCRed text than the title and abstract let on---the failure of the standard n-gram-based methods which work well on born-digital content fail here. Clarifying this in the title and abstract might help ensure the paper reaches the right audience.”
> >
> >
>
>
> We agree that this was an important limitation, which we have addressed by applying our neural de-duplication methods to additional data drawn from C4, a colossal, cleaned version of Common Crawl's web crawl corpus (Raffe et al., 2019). C4 is a highly relevant test case, given its widespread use in training large language models and other deep learning applications. Noise can arise in this corpus for a variety of reasons. To evaluate performance in the presence of varying types of noise, we apply our pre-trained bi-encoder model and hashing to two subsets of C4: RealNews - which consists of around 13 million digital news articles - and all 90,671 patents from Google's online patent database, which is the largest single domain in C4.
>
> The bi-encoder is applied off-the-shelf, with the same hyperparameters as used in the NEWS-COPY analyses. We also use the same definition of a noisy duplicate. For hashing, we found that applying the same parameters resulted in a large number of false positives, detecting 40 times more duplicate pairs than the bi-encoder. This further underscores the brittleness of rule-based methods. Instead, we follow the approach of Borgeaud et al. (2022) for controlling test set leakage in the Retrieval Enhanced Transformer model: hashing with 10-gram collisions and a Jaccard similarity threshold of 0.8. Following GPT-3 (Radford et al., 2019), we ignore clusters above a certain size (using a threshold of 1000 for RealNews and 300 for patents). The bi-encoder detects 27 of these for RealNews and 7 for patents, whereas they do not appear with hashing. These tend to be templates, for things such as medical reports, crime reports, and obituaries. Randomly-selected examples are shown in the Appendix. These might also be undesirable for training and downstream tasks, but we ignore them to be conservative.
>
> Table 4 reports statistics from de-duplicating the training set of RealNews, as well as all patents (which doesn't have a train-test split). The neural method identifies around an order of magnitude more duplicates than hashing. Average cluster size is roughly similar. The appendix provides randomly selected examples of predicted duplicate clusters.
>
> The duplicates predicted by the neural method but not hashing are true positives with reasonably high probability. In a random sample of 50 clusters that are detected by the neural method but not by hashing, 20\% are false positives for RealNews and 4\% are false positives for the patent data. As in NEWS-COPY, many of the false positives consist of updated news stories. When examining duplicates predicted by LSH but not the bi-encoder, 44\% are false positives in RealNews and 16\% are false positives for the patents.
>
> In the RealNews data, news aggregators, which tend to make small edits to digital news stories before releasing them on their own sites (Ingram, 2021), can generate noisy duplicates. In the patents data, one source of duplicates is noisy translation, as patents are routinely filed in multiple countries and non-English language patents were machine-translated into English. OCR noise is also present for older patents. A de-duplicated Google patent dataset could be of direct relevance to researchers working with patent data, which are the backbone of a large literature studying the drivers of innovation.
>
> Finally, Table 5 examines test set leakage on RealNews, using the RealNews development set and the SuperGlue benchmark (Sarlin et al., 2020) as a case study. Exact duplicates are removed in pre-processing. For datasets in SuperGlue that do not primarily consist of news, test set leakage is low. These serve as a placebo, showing that neither method detects duplicates when there is no reason to expect many. In contrast, when examples are drawn from news, the neural method predicts considerable leakage. An analysis of a random sample of (up to) 50 neural predicted duplicates suggests that most are true positives, with the false positive rate ranging from 0\% on 17 duplicates predicted from BoolQ (drawn from Wikipedia) to 16\% on a random sample of 50 duplicates predicted from the RealNews development set. These false positives are often updated articles. In contrast, hashing finds almost no duplicates. The appendix provides examples of predicted duplicates.

---

> ### Author Response · Authors · 2022-11-18
> **Response to reviewer p1KW (part 2)**
>
> (cont from part 1)
> > “Similarly, I was not sure that methods for correcting or compensating for the OCRed text were adequately explored here. The paper does not include concrete examples of what the noisy n-grams look like (this should be added to support the statistics about n-gram noise early in the paper), but it would seem plausible that simple normalization techniques such as removing whitespace and using character n-grams, or mapping each word to a fixed dictionary, might mitigate some of the noise. Techniques like this are left unexplored in the paper.”
>
> We address this important concern in several ways. Examples of texts are now provided in Appendix Sections A.2.1 and A.2.2. We clarify that we remove whitespace and punctuation in pre-processing to reduce noise. With further investigation, we also found that Jaccard similarity led to somewhat better performance of the N-gram overlap method - rather than adjusting the denominator for minimum string length - bringing performance in line with hashing (but still well below neural performance). We also now include ablations in Appendix Table A2 that use character level N-grams, exploring N equal to 10, 15, 20, and 25. Performance is modestly better for hashing and modestly worse for N-gram overlap. Neural methods beat character N-grams by a wide margin.  The character level approach comes at significant additional computational cost. To explore the potential of more aggressive text cleaning to improve the N-gram based methods, we allow SymSpell to correct words up to an edit distance of 5, which results in the overwhelming share of spell-corrected text appearing in the dictionary. As reported in Appendix Table A3, this modestly improves performance for hashing (ARI of 75.9) and N-gram similarity (ARI of 81.1), without changing the overall conclusions.
>
> > “How does the accuracy of the re-ranker change as the bi-encoder threshold for re-ranking varies?”
>
> We now report these results in Appendix Figure A1, documenting the robustness of performance across a range of thresholds. The fact that we use the NEWS-COPY threshold to de-duplicate RealNews and patents, with reasonable results, also suggests that results are reasonably robust to this threshold.
>
> > “Further, the statement about the complexity of the cross-encoder (that it would require $10^{14}$ cross-encodings) seems way too pessimistic given that you could use some kind of temporal constraint to decide which comparisons are worth trying (as I understood it---you have accurate metadata like date and news service).”
>
> We agree that $10^{14}$ cross-encodings would not be required, and have removed this reference. However, with thousands of articles per week in the corpus, the number of cross-encodings is still large. (Unfortunately, we do not have accurate metadata on news wire service, as news wires like the Associated Press used a special ligature that did not always OCR.)
>
> > “As a separate point, I was also unclear on whether some kind of temporal constraint (not allowing clustering across large time spans) would have helped in the experiments reported in Table 2. My guess is not, given the types of errors you describe that that evaluation only considers four distinct days, but it might be helpful to discuss this.”
>
> In the baseline, we pooled the 1930s and 1970s test sets. In appendix Table A4, we now report results separately on the test data from the 1930s and 1970s. Accuracy in the pooled sample is close to the average accuracy when the two decades are separated, as there are few false positive links across decades.
>
> > “Its niche is fairly small (dedupliicating historical texts).”
>
> To address this important concern, we now expand the analysis to the RealNews and patents corpora from C4, providing evidence that the approach works well with modern, digitally generated content as well.
>
> >    “The hand-removed articles that had incorrect bounding boxes—what fraction was this?”
>
> We now clarify that incorrectly merged bounding boxes and news summaries - which summarize multiple news stories in a single article and hence also have no clear cluster with which they are associated - together represent 3.4\% and 3.3\% of the validation and test sets, respectively.
>
> > “The sentence “duplicates are defined…” seems to be missing something about the articles being about the same story, not just from the same service.”
>
> We now clarify this sentence.
>
> > “Congruence labeling was used”--I don’t know what this means, did you measure interannotator agreement? I don't see this in the paper.”
>
> We did mean interannotator agreement, which is now reported. Interannotator agreement on a subset of 8512 pairs was 98.1\% (90.9 Cohen’s Kappa).
>
> We would again like to thank the reviewer for the detailed suggestions, which have been extremely helpful in clarifying and broadening the paper's contribution.

---

> > ### Comment · Reviewer_p1KW · 2022-12-03
> > **Thank you for the response**
> >
> > These changes and additional notes address many of my concerns.  Thank you.

---

### Official Review · Reviewer_szgE · 2022-10-25

**Confidence:** 3
**Correctness:** 2
**Technical Novelty And Significance:** 3
**Empirical Novelty And Significance:** 3
**Recommendation:** 6

**Clarity, Quality, Novelty And Reproducibility:**

The paper is generally well written and I believe has enough substance. Regarding reproducibility, I did not find anything obvious that was missing, but it would be valuable if the authors released their code in addition to the dataset.

**Strength And Weaknesses:**

STRENGTHS
- The new dataset will be a very useful resource for the community. De-duplication is an important step in the data preprocessing pipeline, and this is the first manually annotated dataset to study it that I am aware of.
- Prior work has predominantly used old-school hashing or n-gram based approaches for deduplication, partly because of efficiency constraints. It is great to see a paper studying more advanced approaches, while being mindful about scalability.

WEAKNESSES
- I think that the empirical part of the paper is not convincing. The authors are training neural models on the same distribution that they are testing on, and I am not surprised that this outperforms hashing or n-gram approaches, which do not involve any training. However, this is an artificial setup, as it is not possible to have a labeled training set for de-dupication in the general case. The important question to me is whether the neural approaches also perform better in the wild, or they are just overfitting to this dataset. I understand that this is not trivial to answer, but I believe that the authors could (and should) have done a lot more in this direction. For instance, they could have used the different approaches to deduplicate a crawling corpus, manually evaluate their precision (the percentage of duplicated documents that are truly duplicated), and consider that together with the percentage of predicted duplicate-rate.
- Related to the previous point, all the evaluation in the paper is intrinsic: the authors simply compare the model’s output with the reference annotation according to some automatic metric. If the proposed approach is so much better than hashing and n-gram overlap, it would be good to show its potential in a more practical setup: e.g., show that a language model trained on your deduplicated data is better, revisit data contamination using your deduplication system etc.
- The paper attempts to do some analysis on the behavior and failure cases of different approaches, which is great (e.g. “when neural false positives occur, it is typically in the context of articles that have some repeated content”). However, these seem to be some general observations from the authors without any clear methodology behind and, in addition, no examples are reported, making it difficult for the reader to draw their own conclusions.

**Summary Of The Paper:**

This paper introduces NEWS-COPY, a manually-annotated dataset of 27k documents in the news domain, to study de-duplication. The authors show that two neural approaches (bi-encoder and re-ranking) significantly outperform hashing and n-gram overlap on this dataset, despite the latter being more widely used in the literature. In addition, they show that the bi-encoder approach scales well, de-duplicating a collection of 10 million documents in a few hours using a single GPU.

**Summary Of The Review:**

This paper introduces a very valuable dataset to study de-duplication. This dataset is used to compare neural and hashing/n-gram based approaches to de-duplication, which is an interesting exercise, but more experiments are needed to support that neural approaches are superior in the general case as concluded in the paper. All things considered, I find this to be a borderline paper, perhaps leaning a bit more towards the negative side.

---

> ### Author Response · Authors · 2022-11-18
> **Response to reviewer szgE (part 1)**
>
> > “I think that the empirical part of the paper is not convincing. The authors are training neural models on the same distribution that they are testing on, and I am not surprised that this outperforms hashing or n-gram approaches, which do not involve any training. However, this is an artificial setup, as it is not possible to have a labeled training set for de-dupication in the general case. The important question to me is whether the neural approaches also perform better in the wild, or they are just overfitting to this dataset. I understand that this is not trivial to answer, but I believe that the authors could (and should) have done a lot more in this direction. For instance, they could have used the different approaches to deduplicate a crawling corpus, manually evaluate their precision (the percentage of duplicated documents that are truly duplicated), and consider that together with the percentage of predicted duplicate-rate.”
> >
> >
>
>
> We agree that this was an important limitation, which we have addressed by applying our neural de-duplication methods to additional data drawn from C4, a colossal, cleaned version of Common Crawl's web crawl corpus (Raffe et al., 2019). C4 is a highly relevant test case, given its widespread use in training large language models and other deep learning applications. To evaluate performance in the presence of varying types of noise, we apply our pre-trained bi-encoder model and hashing to two subsets of C4: RealNews - which consists of around 13 million digital news articles - and all 90,671 patents from Google's online patent database, which is the largest single domain in C4.
>
> The bi-encoder is applied off-the-shelf, with the same hyperparameters as used in the NEWS-COPY analyses. We also use the same definition of a noisy duplicate. For hashing, we found that applying the same parameters resulted in a large number of false positives, detecting 40 times more duplicate pairs than the bi-encoder. This further underscores the brittleness of rule-based methods. Instead, we follow the approach of Borgeaud et al. (2022) for controlling test set leakage in the Retrieval Enhanced Transformer model: hashing with 10-gram collisions and a Jaccard similarity threshold of 0.8. Following GPT-3 (Radford et al., 2019), we ignore clusters above a certain size (using a threshold of 1000 for RealNews and 300 for patents). The bi-encoder detects 27 of these for RealNews and 7 for patents, whereas they do not appear with hashing. These tend to be templates, for things such as medical reports, crime reports, and obituaries. Randomly-selected examples are shown in the Appendix. These might also be undesirable for training and downstream tasks, but we ignore them to be conservative.
>
> Table 4 reports statistics from de-duplicating the training set of RealNews, as well as all patents (which doesn't have a train-test split). The neural method identifies around an order of magnitude more duplicates than hashing. Average cluster size is roughly similar. The appendix provides randomly selected examples of predicted duplicate clusters.
>
> The duplicates predicted by the neural method but not hashing are true positives with reasonably high probability. In a random sample of 50 clusters that are detected by the neural method but not by hashing, 20\% are false positives for RealNews and 4\% are false positives for the patent data. As in NEWS-COPY, many of the false positives consist of updated news stories. When examining duplicates predicted by LSH but not the bi-encoder, 44\% are false positives in RealNews and 16\% are false positives for the patents.
>
> In the RealNews data, news aggregators, which tend to make small edits to digital news stories before releasing them on their own sites (Ingram, 2021), can generate noisy duplicates. In the patents data, one source of duplicates is noisy translation, as patents are routinely filed in multiple countries and non-English language patents were machine-translated into English. OCR noise is also present for older patents. A de-duplicated Google patent dataset could be of direct relevance to researchers working with patent data, which are the backbone of a large literature studying the drivers of innovation.

---

> ### Author Response · Authors · 2022-11-18
> **Response to reviewer szgE (part 2)**
>
> (cont from part 1)
>
> > “I think that the empirical part of the paper is not convincing. The authors are training neural models on the same distribution that they are testing on, and I am not surprised that this outperforms hashing or n-gram approaches, which do not involve any training. However, this is an artificial setup, as it is not possible to have a labeled training set for de-dupication in the general case. The important question to me is whether the neural approaches also perform better in the wild, or they are just overfitting to this dataset. I understand that this is not trivial to answer, but I believe that the authors could (and should) have done a lot more in this direction. For instance, they could have used the different approaches to deduplicate a crawling corpus, manually evaluate their precision (the percentage of duplicated documents that are truly duplicated), and consider that together with the percentage of predicted duplicate-rate.”
> >
> >
>
>
> Additionally, Table 5 examines test set leakage on RealNews, using the RealNews development set and the SuperGlue benchmark (Sarlin et al., 2020) as a case study. Exact duplicates are removed in pre-processing. For datasets in SuperGlue that do not primarily consist of news, test set leakage is low. These serve as a placebo, showing that neither method detects duplicates when there is no reason to expect many. In contrast, when examples are drawn from news, the neural method predicts considerable leakage. An analysis of a random sample of (up to) 50 neural predicted duplicates suggests that most are true positives, with the false positive rate ranging from 0\% on 17 duplicates predicted from BoolQ (drawn from Wikipedia) to 16\% on a random sample of 50 duplicates predicted from the RealNews development set. These false positives are often updated articles. In contrast, hashing finds almost no duplicates. The appendix provides examples of predicted duplicates.
>
> > “Related to the previous point, all the evaluation in the paper is intrinsic: the authors simply compare the model’s output with the reference annotation according to some automatic metric. If the proposed approach is so much better than hashing and n-gram similarity, it would be good to show its potential in a more practical setup: e.g., show that a language model trained on your deduplicated data is better, revisit data contamination using your deduplication system etc.”
> >
> >
>
> While we did not have the compute resources to train a language model on a version of C4 de-duplicated with our neural methods, we do now provide an evaluation of test set leakage, as described above. Moreover, Section 2 now provides a more thorough discussion of a large existing literature on the detrimental downstream consequences of duplication in training data. These include regeneration, increased privacy and plagiarism risks, undesired performance if a prompt inadvertently resembles a common duplicate, and reduced performance on downstream tasks ranging from semantic modeling to code understanding to clinical natural language problems. Moreover, the potential for test set leakage is particularly important to address in contexts like language modeling that can consult a massive external database. By releasing the codebase and de-duplicated data, we hope to encourage further investigation of the downstream performance impacts of neural de-duplication.
>
> > “The paper attempts to do some analysis on the behavior and failure cases of different approaches, which is great (e.g. “when neural false positives occur, it is typically in the context of articles that have some repeated content”). However, these seem to be some general observations from the authors without any clear methodology behind and, in addition, no examples are reported, making it difficult for the reader to draw their own conclusions.”
> >
> >
>
> We now randomly sample 50 false positive and 50 false negatives. The supplementary materials provide a systematic coding of the reason for the mistake. Amongst the 50 randomly selected false positive examined, 48 contained articles that were about the same story, but were either from different wire services, or different coverage from the same wire service. 2 were incorrectly labelled. The same errors were made repeatedly. Among 50 randomly selected false negatives examined, in 16 cases, at least one of the articles was very poorly OCR'ed. In 15 cases, one of the articles was missing the first few paragraphs. This case can occur if an article is continued on an interior page. In four cases, one of the articles was heavily truncated, and in another five cases, a significant proportion of the middle of an article was missing. Finally, in four cases, the pair was mislabeled, and in six cases there was no obvious cause of the error. In addition, some randomly selected examples are now provided in Appendix Sections A.2.1 and A.2.2.

---

> ### Author Response · Authors · 2022-11-18
> **Response to reviewer szgE (part 3)**
>
> (cont. from part 2)
>
> > “It would be valuable if the authors released their code in addition to the dataset.”
>
>
> We now clarify that we will release the code as well as all data.
>
>
> We would like to thank the reviewer again for the insightful comments, which have encouraged us to illustrate the utility of neural de-duplication methods across diverse and widely used datasets.

---

> > ### Comment · Reviewer_szgE · 2022-12-02
> > **Response to authors**
> >
> > I would like to thank the authors for their response. It positively addresses my primary concern and I am thus increasing my score. However, I still miss more details about how the qualitative evaluation was conducted. What instructions were annotators given? What criterion was used to determine if two documents are duplicate? Was this blind? In addition, the paper now reports some results on data leakage, but the analysis is rather shallow and the authors do not even study if it has any practical impact (i.e., is accuracy higher in the leaked subset?).

---

> > > ### Author Response · Authors · 2022-12-11
> > > **Response (part 1)**
> > >
> > > Thank you for your additional comments. We are glad to hear that we have addressed your primary concerns. In response to your remaining questions:
> > >
> > > > What instructions were annotators given? What criterion was used to determine if two documents are duplicate? Was this blind?
> > >
> > > Our definition of duplicates for each of the datasets is as follows:
> > >
> > > - NEWS-COPY: For our NEWS-COPY dataset, we defined duplicates as articles that came from the same original wire service source article, regardless of the degree of truncation, abridgement or OCR noise. We started off training annotators with some test labeling samples, which they all labelled. We then organised meetings between annotators to discuss examples where they disagreed and develop a consistent approach to edge cases. We only rolled out labelling for our full dataset when annotators were happy with the coherence of the task, and inter-annotator agreement was over 95% on these test samples. (Note that the inter-annotator agreement reported in the paper was not on these samples on which the annotators were trained, but on a further random sample, once the annotators had been trained.)
> > > - RealNews: we applied the same criteria as we did for our NEWS-COPY dataset, using the same annotators.
> > > - Google Patents: we counted them as duplicates if the same company was using the same language to describe a product.
> > > - SuperGlue: we deduplicated this compared to RealNews, so duplicates were predominantly news articles. Therefore we applied the same criteria as we did for NEWS-COPY, again using the same annotators.
> > >
> > > For the qualitative analysis, we labelled a sub-sample of text clusters pulled out by either the bi-encoder or LSH (but not both), to give an upper bound on false positives. Despite the fact that the bi-encoder extracted far more duplicates than LSH, we found half the number of false positives.
> > >
> > > For RealNews and Google patents, we labelled 50 clusters of texts for each model. The labelling was blind with respect to which model had pulled out the cluster. For SuperGlue, as LSH pulls out very few duplicates, we only analysed those from the bi-encoder, so labelling was not blind.
> > >
> > > For each dataset, we double-labelled a small sample, to check inter-annotator agreement, which is summarised in the table below. Unsurprisingly, inter-annotator agreement was lower on these samples than on NEWS-COPY. This is because these samples were likely to be harder cases (those picked out by only one model, rather than both) and because they were less familiar to labellers (especially in the case of patents).
> > >
> > > |                         Dataset                         | Number of labels | Number double-labelled | Inter-annotator agreement |
> > > |:-------------------------------------------------------:|:----------------:|:----------------------:|:-------------------------:|
> > > | RealNews: duplicates identified by biencoder only       | 50               | 20                     | 65%                       |
> > > | RealNews: duplicates identified by LSH only             | 50               | 20                     | 75%                       |
> > > | Google patents: duplicates identified by biencoder only | 50               | 20                     | 85%                       |
> > > | Google patents: duplicates identified by LSH only       | 50               | 20                     | 75%                       |
> > > | Superglue: duplicates identified by biencoder only      | 71               | 20                     | 100%                      |

---

> > > ### Author Response · Authors · 2022-12-11
> > > **Response (part 2)**
> > >
> > > > In addition, the paper now reports some results on data leakage, but the analysis is rather shallow and the authors do not even study if it has any practical impact (i.e., is accuracy higher in the leaked subset?).
> > >
> > > From the test sets that we considered, we found a particularly high level of training set leakage in ReCoRD, which is only part of SuperGLUE that comprises entirely of news articles.
> > >
> > > ReCoRD is a question-answering task, composed of a passage from a news article and a query with a missing entity, marked @placeholder. The task is for the model to determine which word @placeholder represents. In many cases, where we found a duplicate in RealNews, it was a longer version of the same passage, which included the query sentence, with the correct word in place of @placeholder
> > >
> > > For example, this passage and query pair appears in ReCoRD:
> > >
> > > Passage: *“Minneapolis (CNN) The mayor of Minneapolis said she wants to hear from the officer who fatally shot Justine Ruszczyk. But so far, officer Mohamed Noor has exercised his constitutional right to not speak to state investigators, the Minnesota Bureau of Criminal Apprehension said Tuesday. And, it's not clear if or when he will. "He has a story to tell that no one else can tell," Mayor Betsy Hodges said in a news conference Tuesday. "We can't compel him by law, but I wish that he would make that statement." The news conference capped a day of developments in a case that's raising questions about police training, use of force and body camera policies. The shooting has led newscasts in Australia, where Ruszczyk is originally from. @highlight A county attorney, not a grand jury, will decide whether either of the officers will be charged. @highlight Family shown documents from case ahead of public release, council member says"*
> > >
> > > Query:  *“Those documents were shared with @placeholder's family Tuesday night, she said.”*
> > >
> > > The following near duplicate appears in RealNews:
> > >
> > > _"MINNEAPOLIS - The mayor of Minneapolis said she wants to hear from the officer who fatally shot Justine Ruszczyk. But so far, officer Mohamed Noor has exercised his constitutional right to not speak to state investigators, the Minnesota Bureau of Criminal Apprehension said Tuesday. And, it's not clear if or when he will. The news conference capped a day of developments in a case that's raising questions about police training, use of force and body camera policies. The shooting has led newscasts in Australia, where Ruszczyk is originally from. The department's BCA investigation is expected to last two to four months, said Chuck Laszewski, spokesman for the Hennepin County attorney's office. Once that happens, county attorney Mike Freeman - not a grand jury - will decide whether either of the two officers involved should be charged in Ruszczyk's death. Meanwhile, frustration over the lack of information grows. City Council member Linea Palmisano said some documents from the case would be released online Wednesday morning._ ***Those documents were shared with Ruszczyk's family Tuesday night, she said.*** _[Truncated]”_
> > >
> > > Therefore in these cases of duplicates, the model has often been trained on data containing the answers.
> > >
> > > To evaluate the practical impact of this, we evaluated 100 examples from ReCoRD, zero-shot, with GPT-3, which was trained on Common Crawl (including RealNews). 50 of these are examples that were also in RealNews and 50 that we did not find in RealNews. We prompted GPT-3 with the passage, the query and “Using the information above, what word or words should be part of the final sentence, instead of @placeholder?”. We find that GPT-3 performs substantially better on the examples where a near duplicate was part of the training set, compared to those where it was not.
> > >
> > > |     Dataset    | Accuracy |
> > > |:--------------:|:--------:|
> > > | Duplicates     | 84%      |
> > > | Non-duplicates | 70%      |

---

### Official Review · Reviewer_xk64 · 2022-10-31

**Confidence:** 4
**Correctness:** 3
**Technical Novelty And Significance:** 2
**Empirical Novelty And Significance:** 2
**Recommendation:** 6

**Clarity, Quality, Novelty And Reproducibility:**

The submission is written well and easy to understand. All key steps have been adequately explained with sufficient details. Reproduction of the experimental results should be easy. Dataset creation is based on an interesting premise but it holds only for news corpora. Creating similar datasets for texts from genre other news would not be easy with the proposed approach.

**Strength And Weaknesses:**

Strengths:

1. A new dataset for evaluation of de-duplication methods.
2. Systematic evaluation of neural methods, hashing and n-gram overlap for de-deuplication using the new dataset.
3. Experimental finding that neural methods can give performance gains over the widely used hashing and n-gram methods for de-duplication.

Weaknesses:
1. Experimental evaluation is limited to one dataset and from one domain (news).
2. No novelty in the methodological aspects of representation learning.
3. Impact of the performance gains in deduplication by neural methods on any downstream task is not provided.

**Summary Of The Paper:**

This work addresses the challenge of creating an unbiased evaluation dataset for text de-duplication. It presents one such dataset with 27,210 documents and 122,876 positive duplicate pairs. This dataset was created from news wire data by leveraging the timeliness of news events. It discusses the systematic approach the authors took for creating the dataset.  It uses the dataset for evaluating several de-duplication methods: locality sensitive hashing, N-gram overlap and neural network based and shows that the neural approaches do significantly better than hashing and N-gram methods.

Creating an unbiased dataset for evaluating the efficacy of de-duplication algorithms is a challenge for large document corpora as editorial judgement is not humanly possible for the entire corpus. The paper addresses this challenge by leveraging the time time-sensitivity of news. The basic premise is that in news wire corpora duplicates can occur only within a narrow data range and therefore comprehensive human annotation of duplicates is feasible. Based on this premise, front page article from 973 newspapers on four randomly chosen days in 1930, 1955, and 1974 were reviewed by human annotators to create clusters of duplicate news articles. This resulted in ~27,000 documents ~123,000 positive duplicate pairs. Data from the period 1920-1977 was used as training data.

While n-gram overlap methods seem to be adequate on paper for detecting duplicates, noise in documents generated by OCR of scanned originals can affect higher-order n-grams and reduce the effectiveness of n-gram overlap methods for duplication detection. The work argues that neural approaches are more robust to noise and hence can potentially do better than n-gram methods. It proposes to use two different neural approaches -  1) contrastively trained bi-encoder plus clustering and 2) reranking based on a transformer bi-encoder to measure the pairwise document similarity followed by a cross-encoder to compute cross-attention between the passage and query. A S-BERT MPNET contrastively trained on over a billion sentence pairs is used as the base language model.

Experimental study shows that neural methods do substantially better than both hashing and n-gram overlap methods. Further, neural methods can de-duplicate a 10M article corpus in a matter of hours using a single GPU card. Neural method detected substantially more duplicate pairs than n-gram overlap.

Error analysis of neural methods suggests that false positives are due to articles that have some common content but don't meet the editorial criterion for duplicates. False negatives are due to truncation of the beginning of articles during the digitization process.

Error analysis of n-gram methods suggests that false positives are due to some overlap between non-duplicate articles and false negatives are due to noise added by OCR.




**Summary Of The Review:**

The work discusses the creation of a new dataset for evaluating de-duplication algorithms. The approach taken for dataset creation is interesting but will not scale to genre other than news. Performance gain due to neural methods is very encouraging but as only one dataset from a single domain was used in the evaluation, not much can be said about the utility of the proposed methods for de-duplication in the real world.

---

I have read the reviews by my fellow reviewers and the responses of the authors to the reviews. I thank the authors for their detailed response where they have attempted to address the questions and concerns expressed by me and other reviewers.

Given the very limited innovation in the methodological aspects of the work, I suggest the authors to seriously consider evaluating the relative impact of the proposed neural deduplication method on a few  downstream tasks. This will make the empirical contribution of the work stronger and stand out.  Test set leakage between benchmark datasets and the RealNews corpus is certainly a good step forward in this direction.

Regarding applicability of the proposed method to other corpora and especially from a different domain, the study done on RealNews and Google's online patent database is very encouraging. However, patent database is a relatively small database. Further, false positive rate on RealNews is high (16%) which needs deeper investigation as it can impact the downstream tasks.

Overall I am cautiously positive about this submission and I feel it has the potential to be strengthened significantly and made more useful to the community than it is currently.

---

> ### Author Response · Authors · 2022-11-18
> **Response to Reviewer xk64 (part 1)**
>
> We thank the reviewer for the thoughtful and perceptive comments, which we believe have significantly strengthened the paper. We summarize our responses to these comments and concerns below:
>
>
>
> > “Experimental evaluation is limited to one dataset and from one domain (news)… The approach taken for dataset creation is interesting but will not scale to genre other than news. Performance gain due to neural methods is very encouraging but as only one dataset from a single domain was used in the evaluation, not much can be said about the utility of the proposed methods for de-duplication in the real world.”
> >
> >
>
> We agree that this was an important limitation, which we have addressed by applying our neural de-duplication methods to additional data drawn from C4, a colossal, cleaned version of Common Crawl's web crawl corpus (Raffe et al., 2019). C4 is a highly relevant test case, given its widespread use in training large language models and other deep learning applications. To evaluate performance in the presence of varying types of noise, we apply our pre-trained bi-encoder model and hashing to two subsets of C4: RealNews - which consists of around 13 million digital news articles - and all 90,671 patents from Google's online patent database, which is the largest single domain in C4.
>
> The bi-encoder is applied off-the-shelf, with the same hyperparameters as used in the NEWS-COPY analyses. We also use the same definition of a noisy duplicate. For hashing, we found that applying the same parameters resulted in a large number of false positives, detecting 40 times more duplicate pairs than the bi-encoder. This further underscores the brittleness of rule-based methods. Instead, we follow the approach of Borgeaud et al. (2022) for controlling test set leakage in the Retrieval Enhanced Transformer model: hashing with 10-gram collisions and a Jaccard similarity threshold of 0.8. Following GPT-3 (Radford et al., 2019), we ignore clusters above a certain size (using a threshold of 1000 for RealNews and 300 for patents). The bi-encoder detects 27 of these for RealNews and 7 for patents, whereas they do not appear with hashing. These tend to be templates, for things such as medical reports, crime reports, and obituaries. Randomly-selected examples are shown in the Appendix. These might also be undesirable for training and downstream tasks, but we ignore them to be conservative.
>
> Table 4 reports statistics from de-duplicating the training set of RealNews, as well as all patents (which doesn't have a train-test split). The neural method identifies around an order of magnitude more duplicates than hashing. Average cluster size is roughly similar. The appendix provides randomly selected examples of predicted duplicate clusters.
>
> The duplicates predicted by the neural method but not hashing are true positives with reasonably high probability. In a random sample of 50 clusters that are detected by the neural method but not by hashing, 20\% are false positives for RealNews and 4\% are false positives for the patent data. As in NEWS-COPY, many of the false positives consist of updated news stories. When examining duplicates predicted by LSH but not the bi-encoder, 44\% are false positives in RealNews and 16\% are false positives for the patents.
>
> In the RealNews data, news aggregators, which tend to make small edits to digital news stories before releasing them on their own sites (Ingram, 2021), can generate noisy duplicates. In the patents data, one source of duplicates is noisy translation, as patents are routinely filed in multiple countries and non-English language patents were machine-translated into English. OCR noise is also present for older patents. A de-duplicated Google patent dataset could be of direct relevance to researchers working with patent data, which are the backbone of a large literature studying the drivers of innovation.
>
> Finally, Table 5 examines test set leakage on RealNews, using the RealNews development set and the SuperGlue benchmark (Sarlin et al., 2020) as a case study. Exact duplicates are removed in pre-processing. For datasets in SuperGlue that do not primarily consist of news, test set leakage is low. These serve as a placebo, showing that neither method detects duplicates when there is no reason to expect many. In contrast, when examples are drawn from news, the neural method predicts considerable leakage. An analysis of a random sample of (up to) 50 neural predicted duplicates suggests that most are true positives, with the false positive rate ranging from 0\% on 17 duplicates predicted from BoolQ (drawn from Wikipedia) to 16\% on a random sample of 50 duplicates predicted from the RealNews development set. These false positives are often updated articles. In contrast, hashing finds almost no duplicates. The appendix provides examples of predicted duplicates.

---

> ### Author Response · Authors · 2022-11-18
> **Response to Reviewer xk64 (part 2)**
>
> > “Impact of the performance gains in deduplication by neural methods on any downstream task is not provided.”
>
> We agree that this is an important concern, that we address in two ways. First, we examine test set leakage between benchmark datasets and the RealNews corpus, as described above. Moreover, Section 2 now provides a more thorough discussion of a large existing literature on the detrimental downstream consequences of duplication in training data. These include regeneration, increased privacy and plagiarism risks, undesired performance if a prompt inadvertently resembles a common duplicate, and reduced performance on downstream tasks ranging from semantic modeling to code understanding to clinical natural language problems. Moreover, the potential for test set leakage is particularly important to address in contexts like language models that can consult a massive external database. By releasing the codebase and de-duplicated data, we hope to encourage further investigation of the downstream performance impacts of neural de-duplication.
>
> We would like to thank the reviewer again for the helpful comments, which have encouraged us to show the utility of the approach in other real world applications.

---

### Decision · Program_Chairs · 2023-01-20

**Decision:**

Accept: poster

**Justification For Why Not Higher Score:**

The evaluation in the paper is only based on intrinsic evaluation

**Justification For Why Not Lower Score:**

There is no major concern after the rebuttal.

**Metareview: Summary, Strengths And Weaknesses:**

This paper presents a new data set for news de-duplication and compares several existing approaches, including traditional n-gram-based approaches and neural approaches. The dataset consists of 27K articles and 122K duplicate pairs.

Strengths:
+ In general, the paper is clearly written
+ The research topic is timely and important. The reviews appreciate the creation of the new dataset.
+ There are no major concerns about this paper after the rebuttal

Weaknesses:
- Although the reviewers appreciate the contribution of creating a new dataset, the evaluation provided in the paper is only intrinsic rather than in a practical setup.
- The scope of the paper is narrow. Although it is legitimate for a paper to focus on a specific research problem in a specific domain, it is unclear if the dataset can lead to a general understanding of the ML algorithms and is suitable to publish in a general ML conference like ICLR.



**Note From Pc:**

if the above contains the word "oral" or "spotlight" please see: "oral" presentation means -> notable-top-5% and "spotlight" means -> notable-top-25%. As stated in our emails, we are disassociating presentation type from AC recommendations

**Summary Of Ac-Reviewer Meeting:**

Due to the time differences among reviewers, although reviewers xk64, szgE, and p1KWl provide their availability, only reviewers szgE and p1KW are able to attend the meeting. However, we went over all the reviews and other reviewers have updated their reviews based on the rebuttal.  The discussion is summarized in the final strengths and weaknesses of the papers as listed above.